# XGC-AVis: Towards Audio-Visual Content Understanding with a Multi-Agent Collaborative System

**Yuqin Cao**[1], **Xiongkuo Min**[1], **Yixuan Gao**[1], **Wei Sun**[2],
**Zicheng Zhang**[1,3], **Jinliang Han**[1], **Guangtao Zhai**[1]
[1]Shanghai Jiao Tong University [2]East China Normal University [3]Shanghai AI Laboratory

## Abstract

In this paper, we propose **XGC-AVis**, a multi-agent framework that enhances the audio-video temporal alignment capabilities of multimodal large models (MLLMs) and improves the efficiency of retrieving key video segments through $4$ stages: perception, planning, execution, and reflection. We further introduce **XGC-AVQuiz**, the first benchmark aimed at comprehensively assessing MLLMs' understanding capabilities in both real-world and AI-generated scenarios. XGC-AVQuiz consists of $2,685$ question-answer pairs across 20 tasks, with two key innovations: 1) **AIGC Scenario Expansion:** The benchmark includes $2,232$ videos, comprising $1,102$ professionally generated content (PGC), 753 user-generated content (UGC), and 377 AI-generated content (AIGC). These videos cover 10 major domains and 53 fine-grained categories. 2) **Quality Perception Dimension:** Beyond conventional tasks such as recognition, localization, and reasoning, we introduce a novel quality perception dimension. This requires MLLMs to integrate low-level sensory capabilities with high-level semantic understanding to assess audio-visual quality, synchronization, and coherence. Experimental results on XGC-AVQuiz demonstrate that current MLLMs struggle with quality perception and temporal alignment tasks. XGC-AVis improves these capabilities without requiring additional training, as validated on two benchmarks. The project page is available at: `https://xgc-avis.github.io/XGC-AVis/`

## 1 Introduction

Vision and hearing play crucial roles in human perception and understanding. The human brain can simultaneously perceive multiple modalities, such as text, vision, and audio, and integrate them for joint perception and reasoning. Compared to unimodal inputs, multimodal information enables a more comprehensive understanding and reasoning through cross-modal complementarity. Consequently, enabling multimodal large language models (MLLMs) to efficiently integrate and comprehend multimodal information has become a key direction for future development.

In recent years, MLLMs Li et al. (2024a); Chen et al. (2024); Zhu et al. (2025a); Cheng et al. (2024) have made significant advances, demonstrating remarkable performance in tasks such as dialogue systems, video understanding, and video question answering. Representative models include Chat-GPT Hurst et al. (2024), LLaMA Touvron et al. (2023), Qwen Bai et al. (2023), and DeepSeek families Liu et al. (2024). However, existing models primarily focus on surface-level interactions among text, audio, and visual modalities, and still fall short in understanding the fine-grained cross-modal associations within complex audio-visual (A/V) events. Specifically, there is a lack of modeling for non-speech auditory information, such as object collisions, human actions, and environmental background sounds, which are challenging to semantically align with visual content. Therefore, a growing number of works have begun to explore deeper audio–visual understanding and alignment in large multimodal models Sun et al. (2024); Zhao et al. (2025); Guo et al. (2025). For instance, HumanOmni Zhao et al. (2025) integrates human speech with video for human-centric understanding, SALMONN Sun et al. (2024) advances the comprehension of general audio including music and background sounds, and Dolphin Guo et al. (2025) specifically targets the alignment of audio and visual modalities in the temporal dimension. These efforts highlight the critical importance

of evaluating a model's capacity for modeling non-speech auditory information and fine-grained cross-modal temporal alignment—such as A/V synchronization and temporal localization.

Moreover, current models exhibit limitations in cross-modal temporal alignment. Although MLLMs can process unimodal inputs, they typically underperform in tasks that require precise synchronization between audio and visual signals, such as A/V alignment detection or temporal localization of audio segments corresponding to specific video frames.

Several multimodal benchmarks Gong et al. (2024); Li et al. (2024d); Benchekroun et al. (2023); Zhou et al. (2025) have been proposed to evaluate the understanding capabilities of MLLMs. However, they exhibit three major limitations: **(1) Data source bias:** Existing datasets are primarily collected from user-generated content (UGC) platforms such as YouTube, while lacking professionally generated content (PGC) and AI-generated content (AIGC). PGC offers higher-quality A/V content, such as character close-ups, special effects, and refined visual storytelling. AIGC introduces unrealistic A/V content that doesn't exist in the real world Zhang et al.; Li et al. (2023). **(2) Insufficient task coverage:** Current benchmarks mainly focus on fundamental tasks like A/V recognition and reasoning, while paying limited attention to quality-oriented tasks. Some benchmarks Wang et al. (2025a;b); Zhang et al. (2025) assess MLLMs' visual quality perception but rarely explore A/V quality perception, which evaluates whether MLLMs can perceive the quality or inconsistencies in A/V content, similar to human perception.

In this paper, we introduce **XGC-AVis**, a novel A/V agent system that improves temporal alignment by interweaving video, audio, subtitles, and descriptions. It identifies relevant time segments, ensuring the MLLM's attention on the most significant content. To comprehensively assess XGC-AVis and MLLMs, we present **XGC-AVQuiz**, a holistic benchmark designed to evaluate MLLMs in recognition, localization, quality perception, and reasoning when processing real-world and generative A/V content. XGC-AVQuiz incorporates two key innovations: **(1) Diverse video sources:** XGC-AVQuiz consists of 2232 A/V samples, including 1102 PGC videos, 753 UGC videos, and 377 AIGC videos. These samples effectively address the data diversity limitations of existing benchmarks through the inclusion of real-world and AIGC scenarios. **(2) Multi-level task hierarchy:** The benchmark comprises 2685 carefully constructed question-answer (QA) pairs, covering 4 categories and 20 tasks. The categories are designed with a gradual difficulty progression—from low-level A/V recognition and localization, to high-level A/V reasoning, and comprehensive A/V quality perception. This enables an in-depth assessment of MLLMs' cognitive capabilities across various levels.

We conduct extensive evaluations on a wide range of MLLMs, including both open-source MLLMs, proprietary MLLMs, and multi-agent systems. XGC-AVis demonstrated the best performance on XGC-AVQuiz and Daily-Omni Zhou et al. (2025) benchmarks, showcasing its superior ability in handling complex A/V tasks. The results also reveal significant limitations in current MLLMs' ability to understand A/V content. Specifically, most open-source MLLMs struggle with integrating audio, video, and subtitle information, sometimes failing to outperform Vision-Language Models (VLMs). Additionally, MLLMs face challenges in A/V quality perception tasks, indicating considerable potential for further improvement. Furthermore, MLLMs show weaknesses in A/V tasks involving temporal localization.

## 2 RELATED WORKS

### 2.1 MULTIMODAL LARGE LANGUAGE MODELS

In recent years, MLLMs for audio-visual understanding have made significant progress. Most existing approaches first extract visual embeddings and audio embeddings separately using dedicated visual encoders (e.g., ViT Dosovitskiy et al. (2020), CLIP-ViT Radford et al. (2021)), and audio encoders (e.g., BEATs Chen et al. (2022), HuBERT Hsu et al. (2021)). These embeddings are then concatenated and fed into a large language model for cross-modal reasoning. Some studies Ye et al. (2024); Zhan et al. (2024); Wang et al. (2025c) further incorporate modality-specific decoders, such as speech or music decoders, to enhance the MLLMs' ability to interpret complex audio content. While this paradigm enables accurate recognition of visual objects, actions, speech, and music, it suffers from a key limitation: poor temporal alignment. In this work, we propose the multi-agent system XGC-AVis, which improves the temporal alignment of audio and video events by interweaving video, audio, subtitles, and audio descriptions.

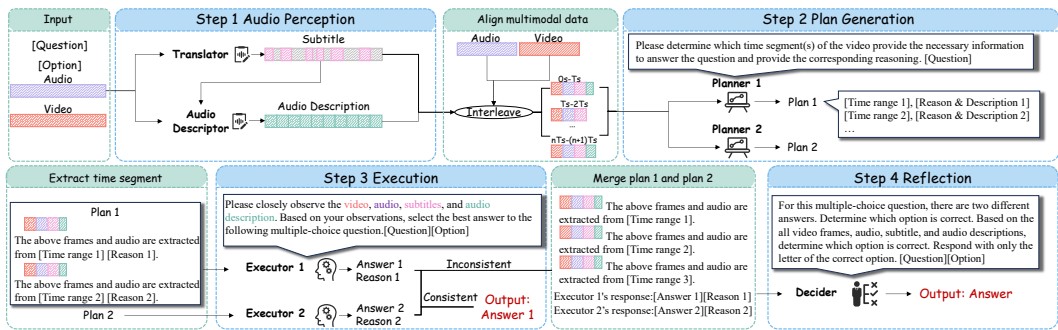

Figure 1: The pipeline of XGC-AVis multi-agent system includes four steps: audio perception, plan generation, execution, and reflection.

## 2.2 MULTI-AGENT SYSTEMS

Recent research has started to explore agent-level and system-level paradigms that mirror complex real-world workflows. For instance, ContextAgent Yang et al. (2025) proposes a context-aware proactive agent for tool invocation, while MAGNET Chowdhury et al. (2025) and OmAgent Zhang et al. (2024) focus on multi-video retrieval and long-video task decomposition, respectively. Similarly, VideoMultiAgents Kugo et al. (2025) utilizes vision and text agents for QA, and Daily-Omni Zhou et al. (2025) employs multi-agent approaches specifically for audio-visual reasoning. In contrast to these works, we propose **XGC-AVis**, which goes beyond simple retrieval to demand fine-grained recognition, localization, and quality perception within complex audio-video content. Unlike previous approaches that often neglect the full potential of the audio modality, XGC-AVis explicitly integrates audio agents and interweaves video, audio, and subtitles to achieve precise spatiotemporal alignment.

## 2.3 MULTIMODAL BENCHMARKS

With the rapid development of MLLMs, a growing number of benchmarks have been proposed. Vision-focused datasets and benchmarks Li et al. (2024c); Zhang et al. (2025); Zhang et al. mainly address perception and understanding tasks involving static images or dynamic videos, while audio-centric datasets and benchmarks Wang et al. (2024); Zhu et al. (2025b) focus on speech, music, and general sound understanding. However, most benchmarks overlook the importance of joint audio-visual perception. Although several benchmarks have been introduced for audio-visual tasks, they still exhibit key limitations. For instance, AV-Odyssey Bench Gong et al. (2024) and OmniBench Li et al. (2024d) focus on static images, while Music-AVQA Li et al. (2022) and AVQA Yang et al. (2022) are domain-specific. Recently, audio-visual benchmarks such as WorldSense Benchekroun et al. (2023) and Daily-Omni Zhou et al. (2025) have been proposed, providing valuable resources for real-world audio-visual question answering. In contrast, our proposed XGC-AVQuiz incorporates PGC, UGC, and AIGC scenarios, explicitly integrates quality perception tasks, and offers a multi-level task hierarchy that enables deeper insights into MLLMs' limitations and guides their advancement toward comprehensive full-modality understanding and perception.

## 3 XGC-AVIS

To enhance the perception and understanding capabilities of MLLMs for multimodal information, we design a multi-agent system named XGC-AVis to assist MLLMs in identifying key time points and temporally aligning video and audio events. As shown in Fig. 1, this process involves four sequential stages: audio perception, plan generation, execution, and reflection. XGC-AVis first segments the audio into equal-length clips. A translator then converts speech into subtitles, while an audio descriptor detects non-speech events. Video frames, audio segments, subtitles, and audio descriptions are interwoven and temporally aligned to form coherent multimodal units. Next, two planners independently analyze these units to identify key time segments relevant to the question. Each planner outputs the relevant time span and reasoning, forming a targeted answering plan that enhances the efficiency of retrieving important video segments. These plans are passed to dedicated executors, each focusing exclusively on its assigned segment and generating an answer along with

Table 1: Comparison with existing audio-visual benchmarks. The table compares existing audio-visual benchmarks based on several key features, including **modality** (A: audio, V: video, I: image), dataset size (**#Videos**, **#QA Pairs**), **annotation** method (A: automatic, M: manual), as well as whether the datasets support **multi-task** QA, include **non-realistic** content, contain **quality perception** questions, and offer diverse **general sound** coverage.

| Benchmarks | Modality | #Video | #QA Pairs | Annotation | Multi-Task | Non-Realistic Content | Quality Perception | General Sound |
|---|---|---|---|---|---|---|---|---|
| AVQA Yang et al. (2022) | A+V | 57,000 | 57,335 | M | ✗ | ✗ | ✗ | ✓ |
| Music-AVQA Li et al. (2022) | A+V | 9,288 | 45,867 | M | ✗ | ✗ | ✗ | ✗ |
| OmniBench Li et al. (2024d) | A+I | ✗ | 1,142 | M | ✓ | ✗ | ✗ | ✓ |
| AV-Odyssey Gong et al. (2024) | A+I | ✗ | 4,555 | M | ✓ | ✗ | ✗ | ✓ |
| WorldSense Benchekroun et al. (2023) | A+V | 1,662 | 3,172 | M | ✓ | ✗ | ✗ | ✓ |
| Daily-Omni Zhou et al. (2025) | A+V | 684 | 1197 | A&M | ✓ | ✗ | ✗ | ✓ |
| **XGC-AVQuiz (Ours)** | A+V | 2232 | 2685 | M | ✓ | ✓ | ✓ | ✓ |

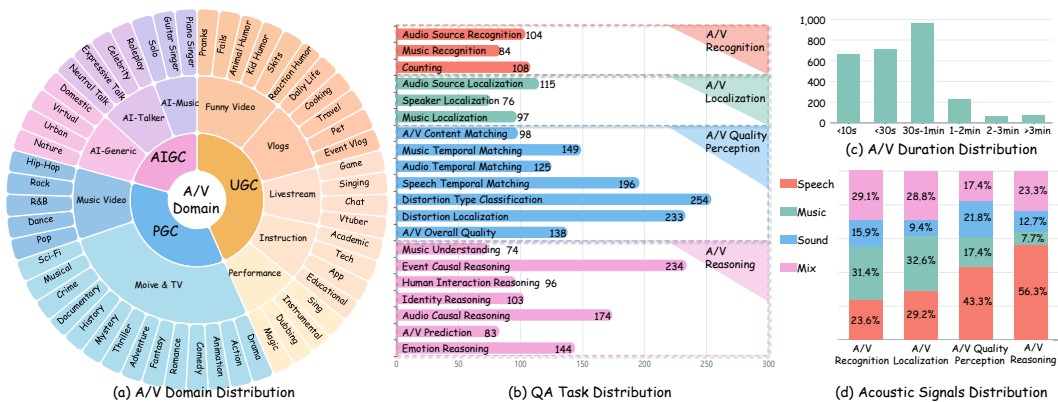

Figure 2: Distribution of videos and question-answer pairs in XGC-AVQuiz.

a reasoning path using associated subtitles, audio descriptions, and contextual cues. If all executors produce the same answer, it is returned directly. In cases of disagreement, the system merges the conflicting plans, answers, and reasoning and forwards them to a decider to derive the final answer. More details about XGC-AVis can be found in Appendix A.1.

In this paper, XGC-AVis employs Deepgram [1] as the translator, r1-aqa Li et al. (2025) as the audio descriptor, Aria Li et al. (2024b) and Qwen2.5-Omni Xu et al. (2025) as Planner 1 and Planner 2, and Gemini 2.0 Flash as Executor 1, Executor 2, and the decider.

## 4 XGC-AVQuiz

In this section, we first provide a brief overview of XGC-AVQuiz and compare it with existing benchmarks in Section 4.1. Section 4.2 describes the video collection and preprocessing pipeline, and Section 4.3 details the QA pair annotation process. Specific QA examples of XGC-AVQuiz can be found in Appendix A.7

### 4.1 BENCHMARK OVERVIEW AND COMPARISON

As illustrated in Fig. 2, XGC-AVQuiz comprises 2, 232 videos, including 1, 102 PGC videos and 753 UGC videos representing real-world scenarios, as well as 377 AIGC videos representing non-realistic scenarios. These videos span 10 major domains and 53 categories. The distribution of video lengths is shown in Fig. 2(c), where 36% of the videos have durations between 30 seconds and 1 minute, with an average duration of 43s. We design a four-level evaluation framework to comprehensively assess A/V understanding. The first level, A/V recognition, focuses on identifying audio and visual events. Next, A/V localization evaluates the model's ability to pinpoint the temporal or spatial location of A/V events. A/V reasoning requires MLLMs to infer relationships and meanings based on A/V events, demonstrating a deeper level of understanding. Finally, A/V quality perception combines low-level sensory analysis with high-level semantic understanding to assess A/V quality. Across these four levels, we further define 20 tasks and collect 2, 685 QA pairs, as shown in Fig. 2(b). Based on the type of audio information required by each QA pair, we categorize

[1]https://developers.deepgram.com/

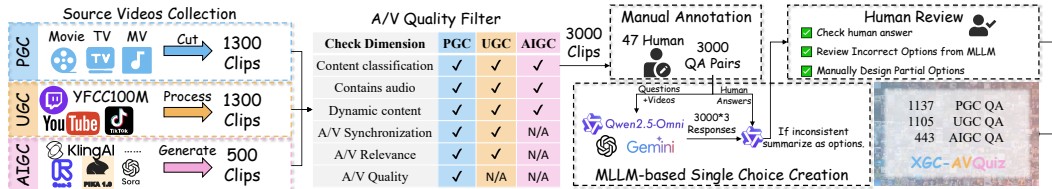

Figure 3: Construction pipeline of the XGC-AVQuiz dataset. (a) Video collection; (b) Quality filtering; (c) Manual annotation; (d) MLLM-based multiple choice creation; (e) Expert review.

them into four types: speech, sound, music, and mixed. The distribution of each audio type across different tasks is illustrated in Fig. 2(d).

As summarized in Table 1, **XGC-AVQuiz** is the first benchmark to comprehensively evaluate the multimodal understanding capabilities of MLLMs across both real-world and non-realistic scenarios. Its key contributions include: (1) **Evaluation of AIGC content**: AIGC content can present A/V scenes that are impossible in the real world. XGC-AVQuiz leverages AIGC videos to evaluate whether MLLMs can semantically and temporally align audio and video, and comprehend non-realistic content. (2) **Emphasis on A/V Quality Perception**: XGC-AVQuiz evaluates MLLMs' ability to perceive A/V quality through low-quality content. This task requires both low-level sensory abilities (e.g., detecting noise, blur, or latency) and high-level understanding (e.g., assessing semantic consistency and temporal synchronization between audio and video). (3) **Broader video coverage**: Compared to existing benchmarks with similar numbers of QA pairs, XGC-AVQuiz includes more videos. This provides broader coverage across diverse video types, scenes, and A/V conditions, enabling a more robust assessment of model generalization and robustness.

## 4.2 DATA COLLECTION AND PREPROCESSING

For PGC videos, we first collected 108 English-language films and TV shows spanning 14 different themes, and trimmed them into 10-minute clips for annotation of QA pairs. Additionally, we downloaded 100 music videos in Korean, English, and Japanese to ensure musical diversity. For UGC videos, we gathered 1,000 short videos from platforms such as YouTube and TikTok, covering game streams, chat sessions, and singing performances. We also collected 100 livestream recordings from Twitch, including game, chat, and singing. Regarding AIGC videos, we sourced AI-generated content from YouTube featuring virtual avatars speaking and singing. To further enrich audio diversity, we also collected AIGC videos with sound from the AIGC sharing platform Kling kli (2024).

To evaluate the A/V quality perception capabilities of MLLMs, we included both high-quality and low-quality A/V content. Low-quality UGC videos were obtained from the UGC A/V quality assessment dataset SJTU-UAV Cao et al. (2023), which contains distortions introduced during user recording. We also simulated livestream-specific glitches, such as audio stuttering, A/V desynchronization, and video freezing, by preprocessing livestream recordings. Low-quality AIGC videos were collected from the AIGC A/V quality assessment dataset AGAVQA-3k Cao et al. (2025), which exhibit issues such as A/V misalignment, semantic inconsistency, and unnatural audio.

After collecting the videos, we first categorized all videos and ensured they contained both audio and dynamic visual content. For high-quality samples, we further verified their A/V synchronization, semantic relevance, and overall A/V quality.

## 4.3 ANNOTATION PROCESS

We recruited 47 professional annotators to create high-quality QA pairs by considering both audio and video information. Annotators were required to generate QA pairs evenly distributed across all 20 task types. All submitted QA pairs were reviewed by domain experts for correctness, clarity, logical consistency, and the necessity of multimodal information. To construct challenging multiple-choice questions, we leveraged MLLMs to generate distractors. Each question and its corresponding video were fed into several MLLMs, and incorrect responses were collected as potential distractors. Finally, professional annotators refined the distractors based on model-generated errors, and all options were reviewed by experts to ensure quality. This annotation framework, combining expert validation with LLM-assisted distractor generation, guarantees the accuracy of QA pairs while increasing the difficulty of multiple-choice questions, thus posing greater challenges to MLLMs.

Table 2: Performance on XGC-AVQuiz across four categories. **Best** and **second-best** results are highlighted. ▲: VLMs. ♦: open-source OLMs. ★: closed-source MLLMs. ♡: multi-agent systems.

| Methods | LLM Size | A/V Recognition | | | A/V Localization | | | A/V Perception | | | A/V Reasoning | | | Average | | |
|---|---|---|---|---|---|---|---|---|---|---|---|---|---|---|---|---|
| | | Vid. | +Aud. | +Sub. | Vid. | +Aud. | +Sub. | Vid. | +Aud. | +Sub. | Vid. | +Aud. | +Sub. | Vid. | +Aud. | +Sub. |
| ▲Qwen2.5-VL | 7B | 42.6 | – | 42.2 | 41.3 | – | 41.3 | 38.6 | – | 38.6 | 41.3 | – | 41.1 | 40.3 | – | 40.1 |
| ▲LLaVA-OneVision | 8B | 47.0 | – | 47.0 | 42.0 | – | 42.7 | 40.2 | – | 41.3 | 47.4 | – | 48.0 | 43.6 | – | 44.4 |
| ▲InternVL2.5 | 8B | 36.8 | – | 37.2 | 42.0 | – | 42.0 | 34.6 | – | 34.2 | 48.1 | – | 48.6 | 40.2 | – | 40.2 |
| ▲InternVL3 | 8B | 48.3 | – | 47.6 | 50.0 | – | **50.3** | 40.4 | – | 40.3 | 52.3 | – | 52.2 | 46.3 | – | 46.2 |
| ♦VideoLLaMA2.1-AV | 7B | 42.6 | 48.0 | 53.0 | 41.0 | 41.7 | 39.2 | 42.5 | 40.7 | 43.1 | 39.9 | 40.2 | 51.7 | 41.5 | 41.4 | 46.7 |
| ♦PandaGPT | 7B | 32.4 | 38.2 | 35.8 | 31.3 | 28.8 | 32.3 | 39.6 | 28.9 | 29.4 | 34.9 | 35.4 | 35.5 | 36.4 | 32.1 | 32.5 |
| ♦GroundingGPT | 7B | 38.9 | 40.5 | 40.9 | 35.8 | 37.8 | 35.8 | 30.1 | 27.2 | 29.5 | 37.6 | 36.2 | 48.8 | 34.2 | 32.8 | 38.0 |
| ♦Bubogpt | 7B | 23.0 | 18.2 | 13.5 | 15.6 | 14.9 | 17.7 | 19.4 | 17.4 | 21.2 | 15.1 | 14.2 | 15.3 | 17.9 | 16.2 | 18.0 |
| ♦Unified-IO-2 | 1B | 30.4 | 31.8 | 30.1 | 29.5 | 28.5 | 29.9 | 38.3 | 35.5 | 34.9 | 28.1 | 28.4 | 26.4 | 33.0 | 31.9 | 30.9 |
| ♦Unified-IO-2 XL | 3B | 34.1 | 35.1 | 35.1 | 24.7 | 25.7 | 25.3 | 28.0 | 28.9 | 27.5 | 27.4 | 29.8 | 27.4 | 28.1 | 29.6 | 28.1 |
| ♦Unified-IO-2 XXL | 8B | 32.4 | 36.5 | 33.4 | 27.1 | 27.4 | 25.3 | 38.7 | 34.2 | 37.3 | 30.4 | 33.0 | 31.1 | 34.0 | 33.3 | 33.9 |
| ♦Video-SALMONN | 7B | 32.4 | 34.5 | 33.8 | 33.7 | 36.5 | 35.4 | 24.7 | 26.2 | 29.2 | 35.5 | 36.0 | 44.7 | 30.2 | 31.5 | 35.6 |
| ♦Qwen2.5-Omni | 7B | 47.6 | 55.4 | 53.7 | 40.6 | 37.5 | 33.0 | 37.1 | **45.1** | 41.8 | 45.0 | 57.9 | 55.1 | 41.3 | 49.8 | 46.7 |
| ★ChatGPT-4o | – | 44.6 | – | 44.9 | 47.2 | – | 47.9 | 23.4 | – | 36.6 | 50.4 | – | 60.1 | 37.4 | – | 46.7 |
| ★Claude 3.7 Sonnet | – | 31.8 | – | 28.7 | 33.7 | – | 29.5 | 15.4 | – | 15.1 | 31.9 | – | 49.6 | 24.8 | – | 29.8 |
| ★Gemini 2.0 Flash | – | 45.9 | 54.4 | **55.7** | 43.4 | 49.3 | 49.3 | 34.3 | 40.2 | 38.1 | 48.5 | 65.9 | **68.1** | 41.3 | **51.4** | 51.3 |
| ♡Daily-Omni | – | 38.9 | | | 25.3 | | | 26.1 | | | 43.7 | | | 33.4 | | |
| ♡XGC-AVis (Ours) | – | **59.5** | | | **51.7** | | | **51.0** | | | **69.3** | | | **58.2** | | |
| *Improvement* | – | *+3.8%* | | | *+1.4%* | | | *+5.9%* | | | *+1.2%* | | | *+6.8%* | | |

Table 3: Performance on Daily-omni across different different question domains and video durations. **Best** and **second-best** results are highlighted. ♦: open-source OLMs. ★: closed-source MLLMs. ♡: multi-agent systems.

| Methods | A/V Event Alignment | Comparative | Context Understanding | Event Sequence | Inference | Reasoning | 30 s Subset | 60 s Subset | Avg |
|---|---|---|---|---|---|---|---|---|---|
| ♦Unified-IO-2XXL (8B) | 44.1 | 51.2 | 38.9 | 40.5 | 57.8 | 61.7 | 46.7 | 48.4 | 47.5 |
| ♦VideoLLaMA2.1-AV (7B) | 35.7 | 35.9 | 35.8 | 31.7 | 40.9 | 34.3 | 38.0 | 31.8 | 35.2 |
| ♦Qwen2.5-Omni (7B) | 25.6 | 31.3 | 26.4 | 25.8 | 35.1 | 29.7 | 26.7 | 30.0 | 28.2 |
| ★ChatGPT-4o | 47.9 | 62.6 | 52.3 | 52.6 | 66.2 | 66.3 | 55.6 | 57.5 | 56.5 |
| ★Gemini 2.0 Flash Lite | 55.0 | 64.9 | 58.0 | 54.3 | 74.0 | 72.0 | 62.4 | 60.0 | 61.3 |
| ★Gemini 2.0 Flash | **62.2** | **73.3** | **63.7** | **63.7** | 76.6 | **75.4** | **67.2** | **68.6** | **67.8** |
| ♡Daily-Omni | 51.7 | 68.7 | 60.1 | 59.3 | **78.6** | 71.4 | 64.0 | 59.3 | 61.8 |
| ♡XGC-AVis (Ours) | **63.5** | **77.1** | **68.4** | **64.4** | **85.1** | **82.3** | **71.6** | **71.5** | **71.5** |
| *Improvement* | *+1.3%* | *+3.8%* | *+4.7%* | *+0.7%* | *+6.5%* | *+6.9%* | *+4.4%* | *+2.9%* | *+3.7%* |

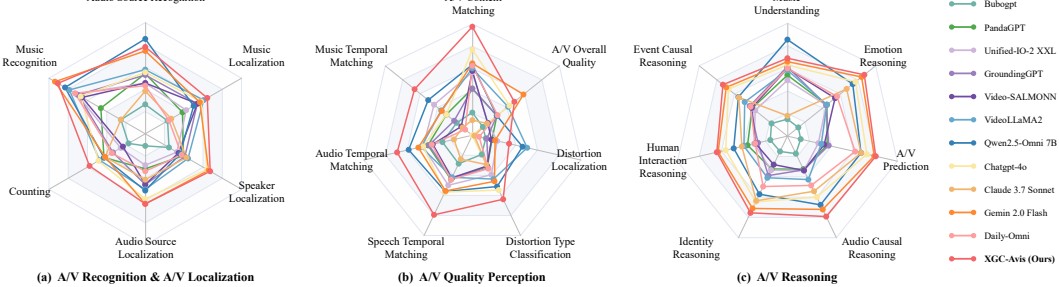

(a) A/V Recognition & A/V Localization    (b) A/V Quality Perception    (c) A/V Reasoning

Figure 4: MLLMs' accuracy over different A/V task categories.

## 5 EXPERIMENT

In this section, we conduct a comprehensive evaluation of recent MLLMs and our proposed XGC-AVis agent on the XGC-AVQuiz benchmark, and further evaluate XGC-AVis on the Daily-Omni benchmark Zhou et al. (2025). In addition, we perform ablation studies on both MLLMs and XGC-AVis to investigate key factors influencing their performance.

### 5.1 EXPERIMENTAL SETTINGS

Our evaluation covers four types of MLLMs: (1) Open-source VLMs: LLaVA-OneVision Li et al. (2024a), InternVL2.5 Chen et al. (2024), and InternVL3 Zhu et al. (2025a). (2) Open-source omni-language models (OLMs): VideoLLaMA2.1-AV Cheng et al. (2024), PandaGPT Su et al. (2023), GroundingGPT Li et al. (2024e), Bubogpt Zhao et al. (2023), Unified-IO-2 Lu et al. (2024), Video-SALMONN Sun et al. (2024), Qwen2.5-Omni Xu et al. (2025). (3) Closed-source MLLMs: GPT-4o

Table 4: Performance on XGC-AVQuiz across different video durations. **Best** and second-best results are highlighted. ♦: open-source OLMs. ★: closed-source MLLMs. ♡: multi-agent systems.

| Methods | LLM Size | < 10s | | | 10s − 30s | | | 30s − 1min | | | 1min − 2min | | | > 2min | | |
|---|---|---|---|---|---|---|---|---|---|---|---|---|---|---|---|---|
| | | Vid. | +Aud. | +Sub. | Vid. | +Aud. | +Sub. | Vid. | +Aud. | +Sub. | Vid. | +Aud. | +Sub. | Vid. | +Aud. | +Sub. |
| ▲LLaVA-OneVision | 8B | 37.8 | – | 37.2 | 47.3 | – | 48.6 | 43.2 | – | 44.6 | 47.3 | – | 48.7 | 49.2 | – | 48.5 |
| ▲InternVL2.5 | 8B | 38.4 | – | 40.5 | 40.5 | – | 39.8 | 37.9 | – | 36.5 | 50.0 | – | 50.4 | 48.5 | – | 50.8 |
| ▲InternVL3 | 8B | 43.4 | – | 42.9 | 49.7 | – | 50.3 | 44.8 | – | 44.0 | 52.7 | – | 53.1 | 43.2 | – | 45.5 |
| ♦VideoLLaMA2.1-AV | 7B | 40.5 | 39.0 | 41.7 | 41.4 | 42.7 | 48.0 | 41.7 | 42.5 | 47.3 | 42.4 | 40.2 | 49.6 | 43.2 | 40.9 | 54.5 |
| ♦PandaGPT | 7B | 30.4 | 30.1 | 30.8 | 38.4 | 39.1 | 38.0 | 39.7 | 27.0 | 28.1 | 31.7 | 35.3 | 38.4 | 38.6 | 36.4 | 33.3 |
| ♦GroundingGPT | 7B | 32.5 | 32.9 | 34.6 | 40.3 | 39.1 | 46.2 | 30.0 | 27.6 | 32.1 | 40.2 | 37.1 | 48.2 | 28.8 | 29.5 | 35.6 |
| ♦Bubogpt | 7B | 15.3 | 14.2 | 15.4 | 21.2 | 18.5 | 16.7 | 17.4 | 16.8 | 21.4 | 18.8 | 12.1 | 14.3 | 15.9 | 15.9 | 19.7 |
| ♦Unified-IO-2 | 1B | 30.7 | 31.0 | 31.3 | 30.5 | 31.5 | 30.5 | 38.4 | 34.8 | 32.4 | 29.5 | 28.6 | 29.9 | 25.8 | 23.5 | 22.7 |
| ♦Unified-IO-2 XL | 3B | 30.5 | 31.0 | 30.2 | 33.9 | 35.0 | 34.3 | 22.9 | 24.1 | 22.7 | 26.3 | 33.9 | 27.7 | 25.8 | 25.8 | 23.5 |
| ♦Unified-IO-2 XXL | 8B | 38.2 | 37.2 | 39.4 | 31.8 | 33.5 | 32.3 | 34.9 | 31.2 | 33.2 | 28.1 | 33.0 | 28.6 | 27.3 | 29.5 | 29.5 |
| ♦Video-SALMONN | 7B | 33.7 | 34.3 | 33.2 | 34.5 | 36.4 | 41.2 | 22.4 | 24.4 | 31.4 | 34.4 | 36.6 | 50.0 | 38.6 | 34.8 | 23.5 |
| ♦Qwen2.5-Omni | 7B | 42.9 | 39.1 | 39.7 | 45.5 | 55.1 | 51.7 | 34.9 | 51.8 | 46.4 | 49.6 | 52.7 | 50.0 | 43.9 | 54.5 | 50.8 |
| ★ChatGPT-4o | – | 43.7 | – | 49.8 | 42.9 | – | 47.6 | 23.1 | – | 37.7 | 53.1 | – | 63.8 | 53.8 | – | 62.1 |
| ★Claude 3.7 Sonnet | – | 30.5 | – | 16.6 | 27.8 | – | 31.2 | 14.7 | – | 29.1 | 33.9 | – | 53.1 | 37.1 | – | 53.8 |
| ★Gemini 2.0 Flash | – | 48.5 | 56.3 | 56.0 | 45.8 | 57.3 | 58.1 | 30.7 | 40.5 | 39.4 | 50.4 | 62.1 | 62.9 | 43.9 | 56.8 | 59.1 |
| ♡Daliy-Omni | – | | 34.7 | | | 36.6 | | | 26.6 | | | 42.0 | | | 43.9 | |
| ♡XGC-AVis (Ours) | – | | **57.1** | | | **62.1** | | | **53.1** | | | **67.4** | | | **63.6** | |
| *Improvement* | – | | *+0.8%* | | | *+4.0%* | | | *+1.3%* | | | *+3.6%* | | | *+1.5%* | |

Table 5: Performance on XGC-AVQuiz across different audio types. **Best** and second-best results are highlighted. ♦: open-source OLMs. ★: closed-source MLLMs. ♡: multi-agent systems.

| Methods | LLM Size | Speech | | | Sound | | | Music | | | Mix | | |
|---|---|---|---|---|---|---|---|---|---|---|---|---|---|
| | | Video | +Audio | +Subtitle | Video | +Audio | +Subtitle | Video | +Audio | +Subtitle | Video | +Audio | +Subtitle |
| ▲LLaVA-OneVision | 8B | 40.7 | – | 39.9 | 38.5 | – | 43.9 | 40.0 | – | 37.4 | 40.7 | – | 40.4 |
| ▲InternVL2.5 | 8B | 45.3 | – | 44.8 | 46.8 | – | 46.3 | 48.0 | – | 49.0 | 46.9 | – | 46.9 |
| ▲InternVL3 | 8B | 42.8 | – | 44.5 | 40.3 | – | 39.9 | 47.3 | – | 47.3 | 44.7 | – | 45.2 |
| ♦VideoLLaMA2.1-AV | 7B | 41.5 | 39.6 | 48.6 | 41.4 | 44.1 | 46.1 | 41.5 | 43.2 | 43.9 | 41.3 | 41.6 | 45.5 |
| ♦PandaGPT | 7B | 38.8 | 31.5 | 32.0 | 34.3 | 34.7 | 33.4 | 33.1 | 32.5 | 32.5 | 35.5 | 31.1 | 32.8 |
| ♦GroundingGPT | 7B | 32.1 | 30.9 | 39.5 | 35.6 | 37.2 | 39.4 | 34.8 | 33.8 | 36.1 | 36.5 | 32.8 | 35.1 |
| ♦Bubogpt | 7B | 18.4 | 17.6 | 18.9 | 12.9 | 15.1 | 12.2 | 24.1 | 15.5 | 19.8 | 16.0 | 14.6 | 19.2 |
| ♦Unified-IO-2 | 1B | 34.6 | 33.8 | 31.6 | 28.1 | 27.4 | 25.4 | 34.2 | 33.8 | 35.1 | 32.8 | 30.1 | 30.7 |
| ♦Unified-IO-2 XL | 3B | 26.2 | 27.7 | 26.7 | 27.6 | 30.3 | 26.9 | 35.9 | 34.8 | 35.7 | 26.1 | 28.5 | 25.6 |
| ♦Unified-IO-2 XXL | 8B | 34.0 | 32.5 | 33.3 | 31.6 | 32.1 | 31.6 | 38.7 | 36.6 | 40.9 | 31.9 | 33.4 | 31.4 |
| ♦Video-SALMONN | 7B | 29.7 | 30.6 | 38.8 | 31.6 | 35.9 | 35.0 | 28.6 | 28.6 | 30.5 | 31.2 | 32.4 | 33.6 |
| ♦Qwen2.5-Omni | 7B | 37.8 | 51.9 | 48.5 | 48.8 | 46.3 | 47.0 | 46.5 | 47.5 | 41.5 | 38.7 | 49.9 | 46.9 |
| ★ChatGPT-4o | – | 32.4 | – | 49.7 | 49.4 | – | 51.4 | 40.9 | – | 43.2 | 35.7 | – | 39.9 |
| ★Claude 3.7 Sonnet | – | 20.4 | – | 42.0 | 25.4 | – | 16.9 | 31.6 | – | 19.4 | 27.7 | – | 23.3 |
| ★Gemini 2.0 Flash | – | 38.7 | 54.1 | 53.3 | 47.7 | 55.7 | 56.1 | 47.3 | 45.4 | 45.2 | 37.2 | 47.7 | 48.7 |
| ♡Daliy-Omni | – | | 36.7 | | | 38.1 | | | 27.5 | | | 27.7 | |
| ♡XGC-AVis (Ours) | – | | **62.7** | | | **57.5** | | | **51.6** | | | **54.8** | |
| *Improvement* | – | | *+8.6%* | | | *+1.4%* | | | *+2.6%* | | | *+4.9%* | |

Hurst et al. (2024), Claude 3.7 Sonnet Anthropic (2024), and Gemini 2.0 Flash DeepMind (2025). (4) Multi-agent systems: Daliy-Omni Zhou et al. (2025) and our XGC-AVis. We further consider three input settings: video only, video with audio, and video with audio and subtitles. In these settings, the questions remain constant while only the input modalities provided to the model vary. More experiment details are in Appendix A.3 and A.4.

## 5.2 MAIN RESULT

Table 2 compares the performance of various MLLMs and multi-agent systems on the four audio-visual tasks in the XGC-AVQuiz benchmark. Our findings offer several key insights into the current state of MLLMs in A/V understanding. **First, open-source VLMs rely solely on visual and textual inputs, which limits their ability to perform multimodal tasks effectively.** In Table 2, we utilize the Deepgram API to transcribe audio into subtitles and feed them to VLMs alongside video. However, this does not significantly improve accuracy compared with video-only input, and VLMs underperform Qwen2.5-Omni in most audio-visual tasks. These results highlight VLMs' inability to leverage subtitle cues and emphasize the necessity of incorporating audio for more comprehensive and accurate A/V understanding.

**Second, most open-source OLMs struggle to integrate audio, video, and subtitle information and fail to surpass VLMs.** Adding audio or subtitles negatively impacts the average accuracy of most open-source OLMs. Only Qwen2.5-Omni and VideoLLaMA2.1-AV effectively leverage audio or subtitle information to achieve higher average accuracy than open-source VLMs. Overall, these

Table 6: Performance on XGC-AVQuiz across difference video content. **Best** and second-best results are highlighted. ♦: open-source OLMs. ★: closed-source MLLMs. ♡: multi-agent systems.

| Methods | LLM Size | PGC | | | UGC | | | AIGC | | |
|---|---|---|---|---|---|---|---|---|---|---|
| | | Video | +Audio | +Subtitle | Video | +Audio | +Subtitle | Video | +Audio | +Subtitle |
| ▲LLaVA-OneVision | 8B | 43.4 | – | 44.0 | 44.1 | – | 45.3 | 42.7 | – | 42.9 |
| ▲InternVL2.5 | 8B | 42.9 | – | 43.7 | 39.2 | – | 37.7 | 35.9 | – | 37.5 |
| ▲InternVL3 | 8B | 49.0 | – | 48.5 | 45.4 | – | 45.4 | 41.8 | – | 42.4 |
| ♦VideoLLaMA2.1-AV | 7B | 37.5 | 38.6 | 48.3 | 44.6 | 44.4 | **46.6** | 43.8 | 41.1 | 42.7 |
| ♦PandaGPT | 7B | 32.8 | 33.2 | 34.6 | 39.9 | 29.2 | 28.9 | 36.6 | 36.6 | 36.1 |
| ♦GroundingGPT | 7B | 35.5 | 34.4 | 43.6 | 32.0 | 31.0 | 32.8 | 35.7 | 33.4 | 36.3 |
| ♦Bubogpt | 7B | 15.7 | 14.3 | 14.2 | 18.7 | 17.4 | 21.3 | 21.4 | 17.8 | 19.4 |
| ♦Unified-IO-2 | 1B | 27.2 | 26.6 | 25.5 | 38.6 | 36.5 | 34.8 | 34.3 | 34.3 | 35.2 |
| ♦Unified-IO-2 XL | 3B | 25.9 | 29.3 | 25.9 | 25.3 | 25.2 | 26.2 | 40.6 | 41.1 | 38.4 |
| ♦Unified-IO-2 XXL | 8B | 27.9 | 30.3 | 28.0 | 36.7 | 33.3 | 36.1 | 42.7 | 41.3 | 43.8 |
| ♦Video-SALMONN | 7B | 33.3 | 34.1 | 41.1 | 26.1 | 27.8 | 30.7 | 32.3 | 34.3 | 33.9 |
| ♦Qwen2.5-Omni | 7B | 43.4 | 55.8 | 54.2 | 35.4 | 44.3 | 38.1 | **51.0** | 48.1 | 48.8 |
| ★ChatGPT-4o | – | 46.6 | – | 55.0 | 27.5 | – | 37.0 | 38.6 | – | 49.7 |
| ★Claude 3.7 Sonnet | – | 28.9 | – | 43.6 | 21.4 | – | 24.1 | 22.3 | – | 8.6 |
| ★Gemini 2.0 Flash | – | 44.4 | 59.3 | **61.6** | 35.7 | 43.7 | 41.7 | 47.6 | 50.6 | 49.2 |
| ♡Daliy-Omni | – | | 41.7 | | | 22.9 | | | 38.1 | |
| ♡**XGC-AVis (Ours)** | – | | **63.8** | | | **54.7** | | | **52.6** | |
| *Improvement* | – | | *+2.2%* | | | *+8.1%* | | | *+1.6%* | |

results indicate that many open-source OLMs remain heavily dependent on visual inputs, and simple concatenation of audio or subtitle signals is insufficient for effective audio-visual integration.

**Third, closed-source MLLMs, such as Gemini 2.0 Flash, show poor performance on quality perception tasks compared to open-source OLMs like Qwen2.5-Omni.** For ChatGPT-4o and Claude 3.7 Sonnet, we follow the official guidelines and use ChatGPT-Audio and Deepgram to generate audio descriptions or subtitles as auxiliary inputs. Results show that for A/V recognition and localization tasks, these inputs offer limited cues. Although Gemini 2.0 Flash achieves the highest average performance among both closed-source and open-source OLMs, its performance on the A/V quality perception task still lags behind the open-source OLM Qwen2.5-Omni, indicating that A/V quality perception remains a key challenge with room for further improvement.

Finally, as shown in Tables 2 and 3, **our proposed XGC-AVis achieves state-of-the-art results, outperforming Gemini 2.0 Flash by 6.8% on XGC-AVQuiz and 3.7% on Daily-Omni in average accuracy.** Moreover, it surpasses the Daily-Omni agent, demonstrating the stronger robustness and transferability of XGC-AVis.

Fig. 4 illustrates MLLMs' accuracy across different A/V task categories. XGC-AVis, which employs Gemini 2.0 Flash as the decider, achieves higher accuracy than Gemini 2.0 Flash on all 20 A/V tasks except the overall A/V quality task. In the A/V recognition and localization categories, most models perform poorly on audio counting. This indicates that although they are capable of capturing semantic information, they struggle to temporally localize A/V events. In the quality perception category, their weakness in distortion localization further suggests difficulty in modeling temporal variations in A/V quality, posing challenges for quality assessment and distortion detection. **These results reveal persistent challenges in temporal localization and variation modeling.** Finally, in the reasoning category, most MLLMs show consistent performance, while Qwen2.5-Omni surpasses Gemini 2.0 Flash in music understanding, indicating a stronger ability to perceive emotional cues.

## 5.3 ABLATION STUDY OF XGC-AVQUIZ

**Impact of Video Duration.** Table 4 illustrates the impact of video durations on MLLMs' accuracy under three input configurations: video-only, video with audio, and video with audio and subtitles. The following insights can be drawn from the results: (1) Our proposed XGC-AVis consistently outperforms other MLLMs, with its largest gain of 4.0% over Gemini 2.0 Flash on 10–30 second videos. (2) As video duration increases, ChatGPT-4o gradually surpasses Gemini 2.0 Flash, showing showing a stronger ability for long content. However, XGC-AVis, built upon Gemini 2.0 Flash, further improves accuracy on videos longer than 30 seconds by leveraging its planner to accurately locate question-relevant segments, thereby enhancing temporal localization. This enables it to surpass both Qwen2.5-Omni and ChatGPT-4o.

Table 7: Ablation studies of XGC-AVis on XGC-AVQuiz. **Best** results are highlighted.

| Methods | Question Type | | | | Video Duration | | | | Audio Type | | | |
|---|---|---|---|---|---|---|---|---|---|---|---|---|
| | A/V Recognition | A/V Localization | A/V Perception | A/V Reasoning | 30s Subset | 1min Subset | 2min Subset | 5min Subset | speech | audio | music | mix |
| Gemini 2.0 Flash | 54.4 | 49.3 | 40.2 | 65.9 | 56.9 | 40.5 | 62.1 | 56.8 | 54.1 | 55.7 | 45.4 | 47.7 |
| XGC-AVis w/o Planner 2 | 57.1 | 48.3 | 43.5 | 65.7 | 53.1 | 49.8 | 61.6 | 60.6 | 56.9 | 51.9 | 49.2 | 49.1 |
| XGC-AVis w/o Planner 1 | 48.0 | 39.9 | 41.4 | 52.5 | 44.9 | 45.5 | 48.7 | 51.5 | 49.3 | 45.9 | 42.4 | 41.1 |
| ♡**XGC-AVis (Ours)** | **59.5** | **51.7** | **51.0** | **69.3** | **59.7** | **53.1** | **67.4** | **63.6** | **62.7** | **57.5** | **51.6** | **54.8** |

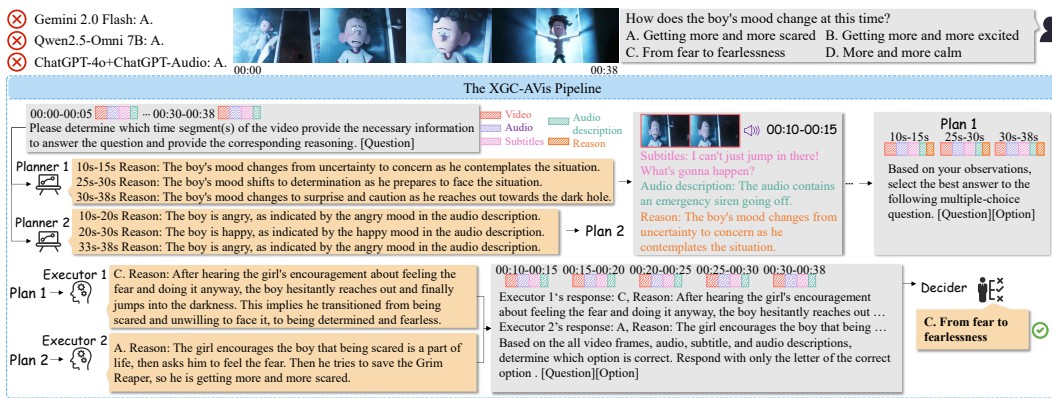

Figure 5: Example of the XGC-AVis response process.

**Impact of Audio Type.** Table 5 compares the impact of four audio types (speech, sound, music, and mix) on MLLMs' performance. XGC-AVis achieves the best performance across all audio types, with notable improvements of 8.6% and 4.9% over Gemini 2.0 Flash on speech and mix, respectively. Although XGC-AVis leverages audio descriptors to aid non-speech sound understanding, significant challenges remain in handling sound and music.

When evaluating both open-source and closed-source MLLMs, we observe: (1) Even for speech, the addition of subtitles may reduce performance, reflecting the challenge MLLMs face in aligning visual content with subtitles. (2) Closed-source MLLMs outperform open-source MLLMs on speech and audio, but lag behind on music and mix. Notably, Gemini 2.0 Flash performs best with video-only input on music, underscoring its limitations in handling music. (3) ChatGPT-4o and Claude 3.7 Sonnet rely on ChatGPT-Audio and Deepgram to acquire audio descriptions and subtitles, respectively. For Claude 3.7 Sonnet, subtitles improve performance only on speech, while ChatGPT-4o benefits from audio descriptions across all audio types. This demonstrates that audio descriptions convey richer acoustic cues, such as tone, emotion, and environmental sounds, that are crucial for multimodal understanding. Unlike subtitles, they capture information beyond textual content, underscoring the importance of comprehensive acoustic signals for effective multimodal comprehension.

**Impact of Video Content.** As shown in Table 6, XGC-AVis achieves the best performance across all three video content types. The largest gain is observed on UGC, where it surpasses the second-best model by 8.1%. This is partly because we introduced low-quality videos into UGC and AIGC to test MLLMs' quality perception. Since VideoLLaMA2.1-AV and Qwen2.5-Omni outperform Gemini 2.0 Flash in quality perception, Gemini 2.0 Flash does not achieve the best results on UGC and AIGC videos. By integrating the insights of Planner 1 and Planner 2, XGC-AVis enhances Gemini 2.0 Flash's ability to perceive quality, thereby improving its overall performance.

## 5.4 ABLATION STUDY OF XGC-AVIS

We evaluate the effectiveness of two planners in XGC-AVis on XGC-AVQuiz, with results shown in Table 7. Gemini 2.0 Flash serves as the baseline model. In XGC-AVis *w/o* Planner 2, the final answer is directly taken from Executor 1, while in XGC-AVis *w/o* Planner 1, it is taken from Executor 2. In both cases, removing either planner leads to a performance drop, demonstrating that the decider's ability to select the correct answer between Executor 1 and Executor 2 significantly improves accuracy and validates the effectiveness of the XGC-AVis architecture. Notably, the improvement is

most pronounced in the A/V quality perception category, providing strong evidence that XGC-AVis effectively enhances MLLMs' performance on quality perception tasks.

## 5.5 CASE STUDY

Fig. 5 presents a case study of how XGC-AVis answers an audio-visual question. In this example, the question is "How does the boy's mood change at this time?" XGC-AVis first concatenates the pre-processed video frames, audio segments, subtitles, and audio descriptions with the question, and feeds them into Planner 1 and Planner 2. Each planner identifies the relevant time segments and provides corresponding reasoning, as shown in their outputs. Based on Planner 1's selected segments, the system constructs Plan 1 and passes it to Executor 1, while Planner 2's outputs form Plan 2, which is processed by Executor 2. If the two executors produce inconsistent answers, XGC-AVis integrates the time spans and reasoning from both plans, along with the candidate answers, and forwards them to the Decider. The Decider then determines the final correct answer.

## 6 CONCLUSION

In this paper, we propose XGC-AVis, a multi-agent system that enhances MLLMs' audio-visual understanding capabilities without additional training. We also introduced XGC-AVQuiz, a novel benchmark designed to comprehensively evaluate MLLMs in both real-world and AI-generated scenarios, with a dedicated focus on quality perception. Experimental results reveal that MLLMs face significant challenges in quality perception and in temporally localizing audio-visual events. Ablation studies demonstrate that XGC-AVis can guide MLLMs to identify question-relevant segments, thereby enhancing temporal localization. Moreover, by integrating decisions from two executors through a decider, XGC-AVis further improves MLLMs' performance in the quality perception category.

## STATMENTS

### ETHICS STATEMENT

This research adheres to the ICLR Code of Ethics. In developing the XGC-AVQuiz benchmark, we invited human annotators to label question-answer pairs. All participants were informed of the task's purpose and their participation was voluntary. No personally identifiable information was collected, and all data was anonymized to ensure privacy and confidentiality. We ensured that the annotation process was free of harm or discomfort for the participants. The annotations were collected solely for academic research, and no sensitive or inappropriate content was involved. Additionally, we took measures to avoid bias in the data collection and ensure fairness in the labeling process. This study complies with relevant ethical standards and legal requirements, with no conflicts of interest or financial sponsors influencing the research. We also ensured that the data usage and participant involvement adhered to ethical guidelines regarding privacy and research integrity.

### REPRODUCIBILITY STATEMENT

We have made significant efforts to ensure the reproducibility of our work. Detailed descriptions of the XGC-AVis framework can be found in Section 3 and Appendix A.1, while Section 4 and Appendix A.3 provide the specifics of the XGC-AVQuiz benchmark. For the models and algorithms used in our experiments, we rely on open-source code and pretrained model weights, or publicly available official APIs, ensuring transparency and reproducibility. Additionally, after the final version of the paper is accepted, we will make the code and dataset publicly available to further facilitate reproducibility and transparency.

### ADDITIONAL LLM STATEMENT

We would like to clarify that LLMs were only used for language polishing and grammar correction in this work. The core research ideas, conceptual framework, experimental design, and data analysis

are entirely the original work of the authors. The authors take full responsibility for the accuracy, completeness, and all statements made in this paper.

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

# A   APPENDIX

## A.1   XGC-AVIS DETAILS

**Audio Perception.** For XGC-AVis, we first use the Deepgram API as a translator to convert the speech in the audio into subtitles. Then, we utilize r1-aqa as an audio descriptor to describe the audio in segments. When the Deepgram API returns subtitles, it also provides the timestamps of each sentence. Therefore, when Deepgram detects speech within an audio segment, we input the detected subtitle along with it into r1-aqa to assist in the speech description.

**Prompt Templates for Audio Descriptor with Speech**

*#User: The subtitle of this audio: [subtitle]. Please describe the background sounds and music. Do not respond using list or dictionary formats. Instead, write your response in full sentences.*

If Deepgram detects that the current audio segment does not contain subtitles, we will prompt r1-aqa that there is no clear speech in the current audio segment, thereby assisting r1-aqa in providing a better description of the audio.

**Prompt Templates for Audio Descriptor without Clear Speech**

*#User: This audio does not contain clear speech. Please the background sounds, emotion and music. Do not respond using list or dictionary formats. Instead, write your response in full sentences.*

**Plan Generation.** After concatenating the video segment, audio segment, subtitle, and audio description, we input them into Planner 1 and Planner 2 to obtain the time segments related to the question, with Aria as Planner 1 and Qwen 2.5-Omni as Planner 2. Aria can only process video and text, while Qwen 2.5-Omni can process video, audio, and text simultaneously.

**Prompt Templates for Planner**

*#User: [video tokens][audio tokens] The above frames and audio are extracted from [Time range]. Subtitle: [Subtitle]. Audio description: [Audio description]...*

*The video lasts for [duration] seconds. Please carefully read the questions related to audio-visual perception, understanding, and reasoning abilities, closely observe the video frames, audio, subtitle, and audio descriptions. Please determine which time segment(s) of the video provide the necessary information to answer the question and provide the corresponding reasoning. Question: [question] No need to answer the question itself, just identify the time range(s) in [start time] to [end time] and provide the corresponding reasoning.*

*Reply me with a structured output in JSON format: "{"time_segments": [{ "start_time": , "end_time": , "reasoning": }...]}" If there is no content related to the question, only answer 'No.'*

**Execution.** Based on the output of the Planner, we extract only the specified time segments and input them into Executor 1 and Executor 2, allowing them to determine the correct option and provide the corresponding reasoning. Both Executor 1 and Executor 2 are based on Gemini 2.0 Flash. If the Planner is unable to identify the video content related to the question, the entire video is assumed to be relevant to the question by default.

**Prompt Templates for Executor**

*#User: [video tokens][audio tokens] The above frames and audio are extracted from [Time range]. Subtitle: [Subtitle]. Audio description: [Audio description]. Video description: [planner's reason]...*

*#Please closely observe the video frames, audio, subtitle, and audio descriptions. Based on your observations, select the best answer to the following multiple-choice question. Respond with only the letter [the range of option letters] of the correct option, followed by 'Reason:' and your reasoning. Question: [Multiple-choice Questions and Options]*

**Reflection.** If the answers from Executor 1 and Executor 2 are consistent, the answer is directly output. When the answers are inconsistent, the responses from Plan 1 and Plan 2 are merged and input into the decider, which will provide the final answer.

Table 8: Comparison of data sources between XGC-AVQuiz and existing benchmarks. We report the distribution of video resolutions and the average shot change frequency.

| Dataset | Resolution Distribution | | | Editing Dynamics |
|---|---|---|---|---|
| | 1080p | 720p | 360p | Shot Change Freq. |
| WorldSense | 0 | 9 | **1653** | 0.18 |
| Daily-Omni | 0 | 0 | 684 | 0.19 |
| **XGC-AVQuiz (Ours)** | **609** | **945** | 1168 | **0.21** |

*#User: [video tokens][audio tokens] The above frames and audio are extracted from [Time range]. Subtitle: [Subtitle]. Audio description: [Audio description]. Video description: [planner's reason]...*

*Executor 1's response: [Answer 1], [Reason 1]. Executor 2's response: [Answer 2], [Reason 2]. For this multiple-choice question, there are two different answers. Based on the all video frames, audio, subtitle, and audio descriptions, determine which option is correct. Respond with only the letter [the range of option letters] of the correct option.*

## A.2 QUANTITATIVE ANALYSIS OF DATA SOURCES

To quantify the distinction between UGC and PGC, we evaluated two key metrics: video resolution and shot change frequency. Specifically, shot changes are detected using PySceneDetect based on pixel-level content discontinuities between consecutive frames. As detailed in Table 8, we conducted a comparative analysis of XGC-AVQuiz against prior benchmarks, specifically WorldSense and Daily-Omni. The results indicate that while existing datasets consist almost exclusively of low-resolution (360p) content, XGC-AVQuiz introduces high-definition standards, containing a substantial volume of 1080p and 720p videos. Furthermore, in terms of editing dynamics, XGC-AVQuiz exhibits a significantly higher shot change frequency compared to WorldSense and Daily-Omni. This quantitative evidence validates that although prior benchmarks host diverse content, they have predominantly relied on static, low-fidelity data. In contrast, XGC-AVQuiz fills this critical gap by providing professionally produced, high-fidelity, and dynamically edited content.

## A.3 MLLM EXPERIMENT DETAILS

In the XGC-AVQuiz benchmark, each question contains 2 to 6 carefully designed options, with only one correct answer. The correct and incorrect answers are shuffled during the evaluation process. During the evaluation of MLLMs, we uniformly select 15 frames from each video and provide the complete audio segment as input.

**Prompt Template for MLLM Evaluation**

*#User: [video tokens][audio tokens]*

*These are the frames of the video and the corresponding audio. Please select the best answer to the following multiple-choice question based on the video. Respond with only the letter [the range of option letters (e.g., A, B, C, D).] of the correct option.*

*Question: [Multiple-choice Questions and Options]*

All MLLMs are tested in a zero-shot setting. For open-source MLLMs, we conducted tests by downloading the official default parameters and running the tests on two A100 GPUs with 160 GB of memory. Closed-source MLLMs are evaluated via official APIs to ensure the reproducibility of the results. Apart from ChatGPT-4o, we used the official Deepgram API to convert the audio content of each video into text, which is then used as subtitles for testing MLLM performance.

**Prompt Template for MLLM Evaluation with Subtitles**

*#User: [video tokens][audio tokens]*

*These are the frames of the video and the corresponding audio. The subtitle of this video: [subtitle]. Please select the best answer to the following multiple-choice question based on the video. Respond with only the letter [the range of option letters (e.g., A, B, C, D).] of the correct option.*

Table 9: Ablation study on the effectiveness of key video segment retrieval. We compare our method against using the full video input (*w/o* key segments) across four standard metrics.

| Method | A/V Recognition | A/V Localization | A/V Perception | A/V Reasoning |
|---|---|---|---|---|
| XGC-Avis *w/o* key video segments | 56.1 | 50.1 | 45.1 | 68.3 |
| XGC-AVis (Ours) | 59.5 | 51.7 | 51.0 | 69.3 |

For ChatGPT-4o, the official ChatGPT-Audio is used, and the description of the audio by ChatGPT-Audio is input as subtitles to assist ChatGPT-4o in understanding the audio-visual content.

**Prompt Template for ChatGPT-Audio**

*#User:Listen to the audio carefully. Describe the sounds in the audio, convert the spoken words into accurate subtitles. [audio tokens]*

*Question: [Multiple-choice Questions and Options]*

### A.4  DETAILS ON LLM-ASSISTED EVALUATION

During the experiment, we found that some MLLMs models output answers in a format that does not meet the requirements. For example, the MLLMs would not respond with just the letter of the option, but would instead add additional explanatory words. To address this, we adopted an LLM-assisted evaluation method. We input the question, options, correct answer, and the MLLM's response into an LLM to evaluate the accuracy of the answer. We used Qwen-plus to assist in judging the accuracy of MLLM responses. To reduce the inherent variability of large language models, where identical prompts may produce uncertain responses, we employed a five-round voting strategy. For each question-answer pair, we sent the prompt defined in the template below five times and determined the correctness of the answer based on the majority, selecting the result that appeared three or more times. When the response from the LLM-assisted evaluation does not meet the requirements, we manually verify the accuracy of the MLLM's answer.

**Prompt Templates for LLM Evaluation of MLLMs' Responses.**

*#User: Given the question [multiple-choice question and options, the correct answer is the option [correct answer]. The respondent's answer is [MLLM's answer]. Determine if the respondent's answer is correct (1) or incorrect (0). If uncertain, also provide 0. Only return the result as a single digit.]*

### A.5  MORE EXPERIMENTS

#### A.5.1  EFFECTIVENESS OF KEY SEGMENT RETRIEVAL

To validate the efficiency of our key segment retrieval, we conducted a comparative analysis by replacing the key segments selected by XGC-AVis with the entire video input. As presented in the Table 9, utilizing the full video actually led to performance decreases across all four categories. This indicates that our selection mechanism effectively filters irrelevant noise. Furthermore, we conducted a user study with 10 participants who answered questions based solely on the key segments identified by our planners, without viewing the full video. Participants achieved high accuracy rates of 87% for segments selected by Planner 1 and 84% for Planner 2. Collectively, these experiments demonstrate that XGC-AVis significantly improves the efficiency of retrieving key video segments by accurately isolating the most critical content.

#### A.5.2  EXPERIMENTS ON WORLDSENSE

To further evaluate the generalization capabilities of our approach, we conducted experiments on the WorldSense benchmark. As summarized in Table 10, XGC-AVis achieves an average accuracy of 51.1%, consistently outperforming the strong baseline Gemini 2.0 Flash as well as other state-of-the-art models such as Qwen2-VL and Qwen 2.5 Omni, validating the robust generalization ability of XGC-AVis in handling open-world audio-visual scenarios.

Table 10: Performance comparison on the WorldSense benchmark. Best results are in bold. ♦: open-source OLMs. ★: closed-source MLLMs. ♡: multi-agent systems.

| Method | Tech & Sci | Culture & Pol | Daily Life | Film & TV | Performance | Games | Sports | Music | Average |
|---|---|---|---|---|---|---|---|---|---|
| ▲LLaVA-OneVision | 38.9 | 38.9 | 36.3 | 37.6 | 37.8 | 37.9 | 36.3 | 39.1 | 37.7 |
| ▲InternVL2.5 | 43.7 | 40.9 | 34.6 | 39.7 | 37.8 | 36.2 | 39.4 | 41.1 | 39.1 |
| ▲Qwen2-VL | 33.5 | 29.0 | 28.4 | 33.6 | 30.3 | 32.3 | 34.7 | 38.5 | 32.4 |
| ♦Unified-IO-2 | 19.3 | 22.8 | 23.1 | 25.6 | 25.8 | 24.1 | 22.9 | 25.3 | 23.3 |
| ♦Unified-IO-2 XL | 26.5 | 24.4 | 22.5 | 23.5 | 24.7 | 28.0 | 25.7 | 24.2 | 24.7 |
| ♦Unified-IO-2 XXL | 27.1 | 31.7 | 23.9 | 23.7 | 25.5 | 23.7 | 25.7 | 27.3 | 25.9 |
| ♦Video-SALMONN | 57.1 | 54.4 | 48.9 | 50.9 | 49.1 | 51.1 | 44.9 | 51.0 | 50.9 |
| ♦Qwen2.5-Omni | 47.8 | 49.8 | 43.6 | 43.8 | 48.3 | 39.1 | 43.5 | 47.3 | 45.4 |
| ★GPT-4o | 48.0 | 44.0 | 38.3 | 43.5 | 41.9 | 41.2 | 42.6 | 42.7 | 42.6 |
| ★Gemini 2.0 Flash | 54.1 | 50.1 | 50.4 | 49.0 | 52.3 | 48.5 | 45.9 | 47.9 | 49.9 |
| ♡**XGC-AVis (Ours)** | **54.1** | **50.2** | **51.7** | **50.1** | **54.7** | **50.2** | **47.2** | **50.5** | **51.1** |

Table 11: Ablation studys of XGC-AVis on XGC-AVQuiz. We compare the impact of removing the interleaving step and replacing the backbone model with Qwen2.5 Omni.

| Method | A/V Recognition | A/V Localization | A/V Perception | A/V Reasoning | Average |
|---|---|---|---|---|---|
| Qwen2.5 Omni (Baseline) | 55.4 | 37.5 | 45.1 | 57.9 | 49.8 |
| XGC-AVis (w/ Qwen2.5 Omni) | 55.7 | 38.1 | 47.1 | 58.1 | 50.8 |
| XGC-AVis (w/o Interleaving) | 57.2 | 47.1 | 45.1 | 67.5 | 54.2 |
| **XGC-AVis (Ours)** | **59.5** | **51.7** | **51.0** | **69.3** | **58.2** |

### A.5.3 IMPACT OF INTERLEAVING

Recent advancements in MLLMs have demonstrated that interleaving visual tokens with textual descriptions (or timestamps) is a highly effective strategy for bridging the semantic gap. Aria Li et al. (2024b) enhances temporal understanding by explicitly interleaving timestamp text with video frames, proving that text-visual interleaving effectively aids the model in grounding temporal information. Flamingo Alayrac et al. (2022), VILA Lin et al. (2024) and MM1 McKinzie et al. (2024) demonstrate that training on interleaved image-text data enables models to perform complex reasoning across modalities. These works highlight that the architecture of interleaving data inputs is crucial for unlocking few-shot learning and in-context reasoning capabilities in MLLMs.

To validate the efficacy of our core temporal alignment mechanism, we conducted an ablation study by removing the specific "Interleave" step from our pipeline. As shown in the third row of Table 11, eliminating this step results in a significant decline in average performance, dropping from $58.2\%$ to $54.2\%$. This substantial gap confirms that our strategy of explicitly interweaving video frames, audio segments, and textual cues is essential for achieving fine-grained cross-modal reasoning.

### A.5.4 GENERALIZABILITY TO OPEN-SOURCE MODELS

We further assessed the architectural versatility of XGC-AVis by replacing the proprietary Gemini 2.0 Flash with the open-source Qwen2.5-Omni model for both the executors and decider. As presented in Table 11, the resulting system achieves an accuracy of $50.8\%$, which still outperforms the Qwen2.5-Omni baseline. This improvement demonstrates that the performance gains are not solely derived from the strength of the underlying foundation model. Instead, they are primarily driven by the structural advantages of the XGC-AVis framework itself, proving its feasibility and effectiveness even in open-source environments.

### A.5.5 EVALUATION OF TEMPORAL LOCALIZATION ACCURACY

While our primary evaluation metrics focus on cross-modal content association, we further assess precise time-alignment capabilities using Temporal Localization Accuracy (TLA). Specifically, we filtered a subset of questions that require numerical timestamp answers to calculate this metric. XGC-AVis achieves the highest TLA of $28.6\%$, surpassing the strong closed-source baseline Gemini 2.0 Flash ($26.4\%$) and Qwen 2.5 Omni ($24.1\%$). This result confirms that our interleaved architecture effectively enhances fine-grained temporal grounding.

Table 12: Performance comparison across different input configurations: Video-only (V), Audio-only (A), and Video+Audio (V+A). The results demonstrate that combined multimodal input consistently yields the best performance.

| Method | A/V Recognition | | | A/V Localization | | | A/V Perception | | | A/V Reasoning | | | Average | | |
|---|---|---|---|---|---|---|---|---|---|---|---|---|---|---|---|
| | V | A | V+A | V | A | V+A | V | A | V+A | V | A | V+A | V | A | V+A |
| ♦PandaGPT | 32.4 | 37.8 | 35.8 | 31.3 | 30.9 | 32.3 | 39.6 | 31.6 | 29.4 | 34.9 | 37.0 | 35.5 | 36.4 | 34.0 | 32.5 |
| ♦GroundingGPT | 38.9 | 38.2 | 40.9 | 35.8 | 28.8 | 35.8 | 30.1 | 29.1 | 29.5 | 37.6 | 32.5 | 48.8 | 34.2 | 31.2 | 38.0 |
| ♦BuboGPT | 23.0 | 27.7 | 13.5 | 15.6 | 26.0 | 17.7 | 19.4 | 34.2 | 21.2 | 15.1 | 25.0 | 15.3 | 17.9 | 29.5 | 18.0 |
| ♦Unified-IO-2 | 30.4 | 31.8 | 30.1 | 29.5 | 21.2 | 29.9 | 38.3 | 39.6 | 34.9 | 28.1 | 29.2 | 26.4 | 33.0 | 33.3 | 30.9 |
| ♦Unified-IO-2 XL | 34.1 | 35.8 | 35.1 | 24.7 | 26.7 | 25.3 | 28.0 | 28.2 | 27.5 | 27.4 | 27.4 | 27.4 | 28.1 | 28.6 | 28.1 |
| ♦Unified-IO-2 XXL | 32.4 | 42.6 | 33.4 | 27.1 | 30.9 | 29.5 | 38.7 | 40.5 | 37.3 | 30.4 | 33.0 | 31.1 | 34.0 | 37.2 | 33.9 |
| ♦Qwen2.5-Omni | 47.6 | 52.4 | 53.7 | 40.6 | 39.6 | 33.0 | 37.1 | 44.8 | 41.8 | 45.0 | 49.3 | 55.1 | 41.3 | 46.6 | 46.7 |
| ★Gemini 2.0 Flash | 45.9 | 51.4 | **55.7** | 43.4 | 39.2 | **49.3** | 34.3 | 36.0 | **38.1** | 48.5 | 60.6 | **68.1** | 41.3 | 46.4 | **51.4** |

Table 13: Comparison of average inference time across varying video lengths.

| Method | 10s | 30s | 1min | 5min |
|---|---|---|---|---|
| Gemini 2.0 Flash | 21.76s | 22.72s | 22.59s | 23.57s |
| XGC-AVis (Ours) | 60.14s | 62.21s | 64.21s | 65.14s |

### A.5.6 IMPACT OF SINGLE INPUT MODALITY

To verify that our benchmark necessitates multimodal reasoning rather than being solvable via a single modality, we evaluated performance across three input configurations: video-only input (V), audio-only input (A), and audio-visual input (V+A). As shown in Table 12, existing models generally struggle when relying on a single modality. The results indicate that the XGC-AVQuiz benchmark cannot be effectively solved through visual or audio cues alone. Notably, for advanced models like Gemini 2.0 Flash and Qwen2.5-Omni, the combined V+A input consistently yields the highest average performance compared to Video-only or Audio-only settings. This performance gap confirms the validity and necessity of the multimodal design inherent to XGC-AVis.

## A.6 COMPUTATIONAL EFFICIENCY AND SCALABILITY

### A.6.1 COMPUTATIONAL COST

As shown in Table 13, we evaluated the average inference time across varying video lengths on two NVIDIA RTX 6000 (96GB). For the baseline, Gemini 2.0 Flash employs a fixed 15-frame input strategy, resulting in constant inference time. For XGC-AVis, we report the full pipeline execution time assume the two executors are consistently distinct. Although XGC-AVis exhibits higher latency due to its multi-step architecture, the inference time remains highly stable as video length increases. Processing a 5 minute video takes only 5s longer than a 10 second video, demonstrating efficient scaling without exponential cost. We regard this additional time as a necessary trade-off, prioritizing performance to enable the fine-grained cross-modal reasoning that monolithic MLLMs lack.

### A.6.2 DISCUSSION OF SCALABILITY

We address scalability concerns by highlighting that XGC-AVis employs parallel execution via multiprocessing for both Plan Generation and Execution steps. This design ensures that scaling the number of agents introduces minimal latency overhead. Specifically, increasing the number of planners from $N = 2$ to $N = 4$ doubled the GPU memory requirement (120GB $\rightarrow$ 240GB) but increased the average latency for a 1 minute video by only 2 seconds (64.21s $\rightarrow$ 66.33s). Regarding input scalability, to prevent runtime explosion with longer videos, we implement a maximum cap of 80 frames. This strategy imposes an effective upper bound on runtime regardless of video duration, thereby ensuring consistent efficiency.

## A.7 QA EXAMPLES FROM EACH TASK

XGC-AVQuiz encompasses 4 A/V categories and 20 A/V tasks. To provide a clearer understanding of each task type, we present one QA example for each task.

### A.7.1 A/V RECOGNITION CATEGORY

Fig. 6 illustrates 3 representative tasks from the A/V recognition category: audio source recognition, music recognition, and counting. The audio source recognition task evaluates whether MLLMs can correctly identify the types of sounds and their corresponding sources in a video. The music recognition task focuses on assessing MLLMs' ability to recognize the genre or composition of music within the video. The counting task challenges MLLMs to determine the number of occurrences of a specific sound event.

### A.7.2 A/V LOCALIZATION CATEGORY

Fig. 7 illustrates 3 representative tasks from the A/V localization category: audio source localization, speaker localization, and music localization. The audio source localization task evaluates MLLMs' ability to identify the position of non-speech, non-music sound sources in the video, as well as track variations in the sound source's volume over time. The speaker localization task tests MLLMs' capability to determine the positions and the temporal sequence of speakers in the video. The music localization task requires the model to detect the moments when music enters the scene and identify the time points when the music changes.

### A.7.3 A/V QUALITY PERCEPTION CATEGORY

Fig. 8 and Fig. 9 illustrates 7 representative tasks from the A/V quality perception category: A/V content matching, music temporal matching, audio temporal matching, speech temporal matching, distortion type classification, distortion localization and A/V overall quality. The A/V content matching task tests whether MLLMs can determine the degree of alignment between the video and audio content in AIGC audio-visual content. The music temporal matching task assesses whether MLLMs can perceive the synchronization between music events and corresponding visual content, such as whether the singer's lip movements align with the music. The audio temporal matching task evaluates the synchronization of non-speech and non-music audio events with the video content in terms of timing. The speech temporal matching task requires MLLMs to assess the temporal alignment between speech and video content, such as whether the speaker's lip movements correspond to the spoken words. The distortion type classification task classifies the type of distortion present in a given audio-visual content, such as noise, blur, or compression artifacts. The distortion localization task identifies the specific regions in the video where distortions occur, localizing them both spatially and temporally. The A/V overall quality task evaluates the overall perceptual quality of the audio-visual content, combining both audio and video attributes into a single quality score.

### A.7.4 A/V REASONING CATEGORY

Fig. 10 and Fig. 11 illustrates 7 representative tasks from the A/V reasoning category: music understanding, event causal reasoning, human interaction reasoning, identity reasoning, audio causal reasoning, A/V prediction, and emotion reasoning. The music understanding task requires MLLMs to comprehend the meaning conveyed by the music in the video or determine what aspects of the video the music aims to highlight. The event causal reasoning task challenges MLLMs to reason about the storyline or the causes behind specific events based on the audio and video cues. The human interaction reasoning task asks MLLMs to infer the reasons behind the actions of characters in the video, combining both audio and video events. The identity reasoning task requires MLLMs to deduce the identity of characters or the relationships between two characters based on the audio-visual information. The audio causal reasoning task directs MLLMs to focus on audio events, inferring the causes and effects of specific sounds within the video. The A/V prediction task requires MLLMs to predict future events, behaviors, or sounds in the video by combining audio and visual information. The emotion reasoning task involves judging emotions, tracking emotional changes, and understanding the causes behind these emotional shifts in the video content.

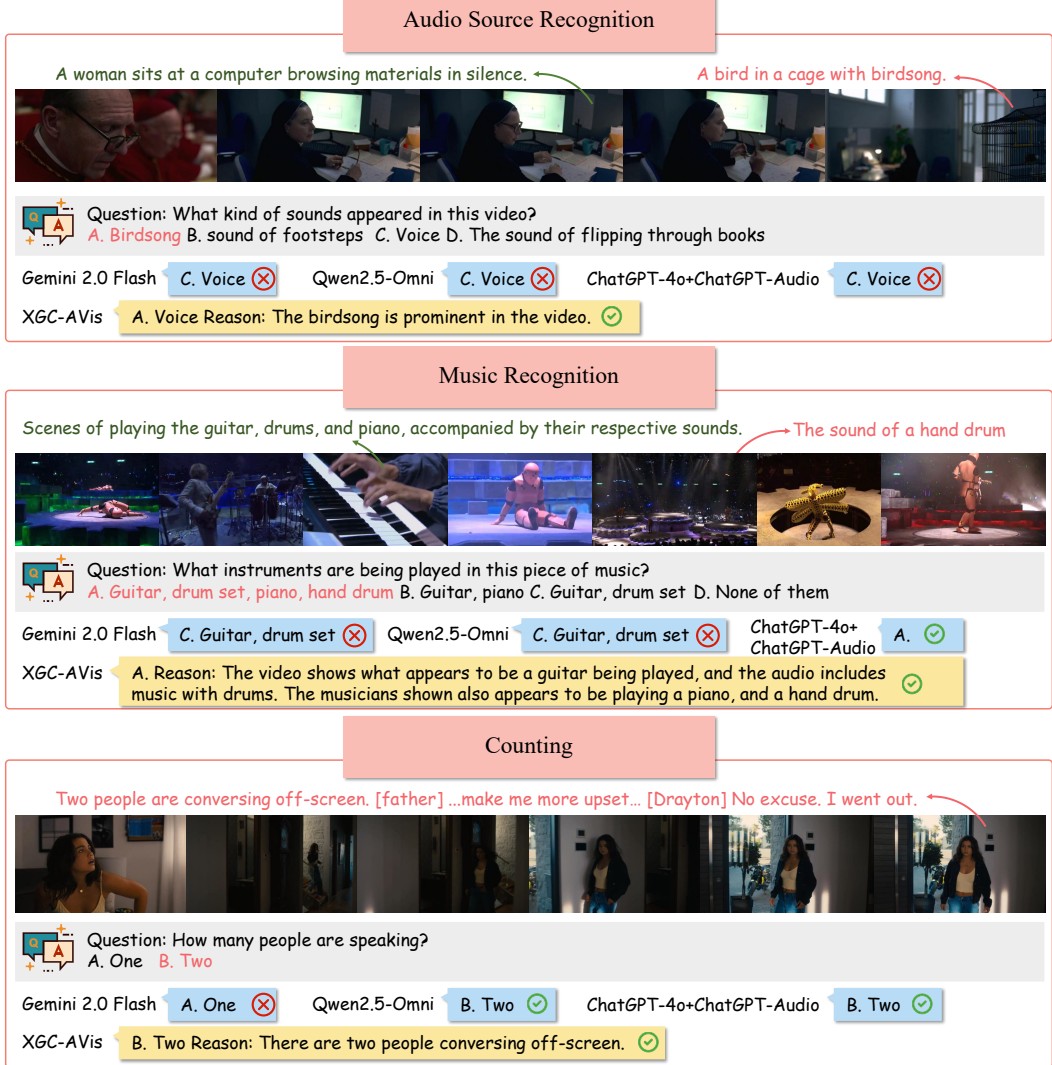

Figure 6: Examples of MLLM and AVis responses in the A/V recognition category, including audio source recognition, music recognition, and counting tasks.

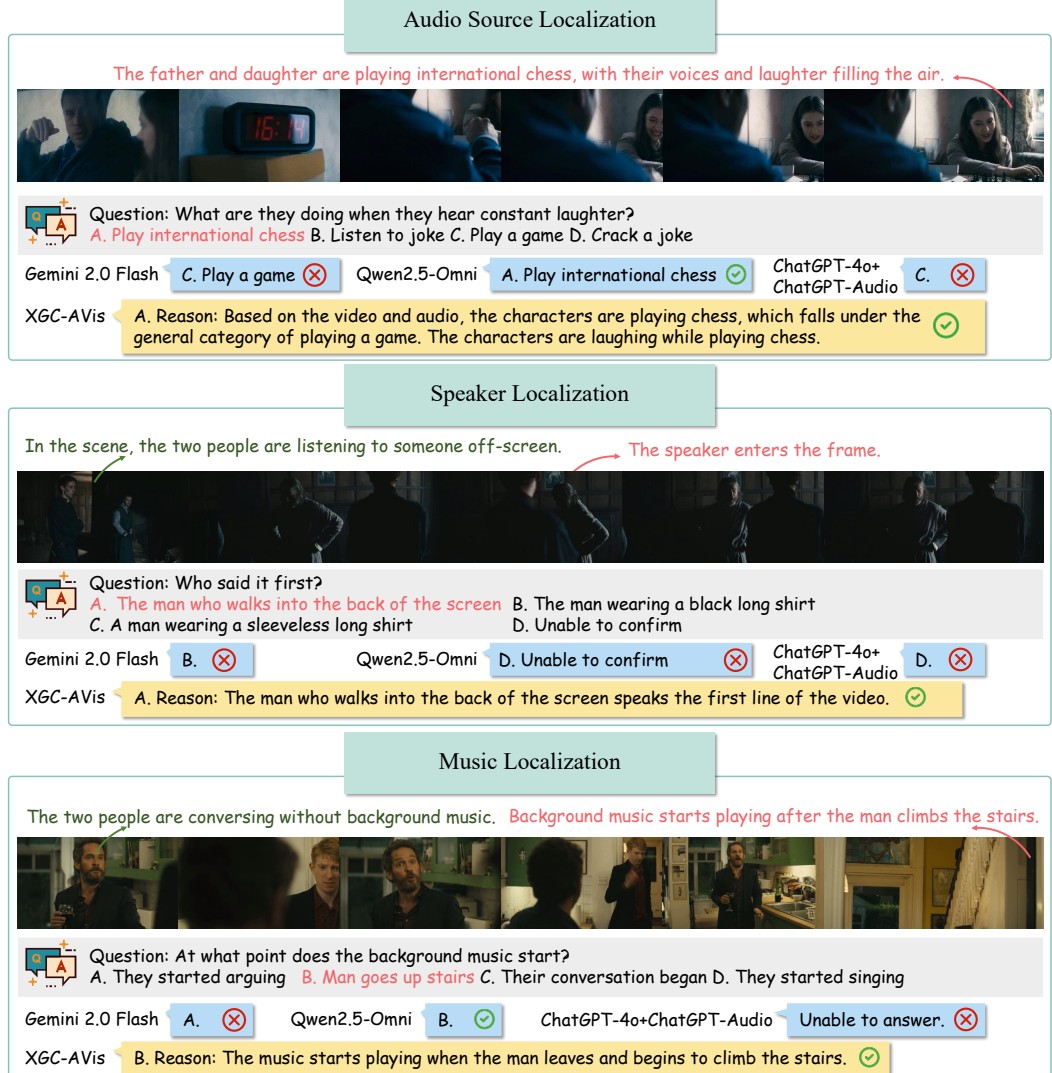

Figure 7: Examples of MLLM and XGC-AVis responses in the A/V localization category, including audio source localization, speaker localization and muisc localization.

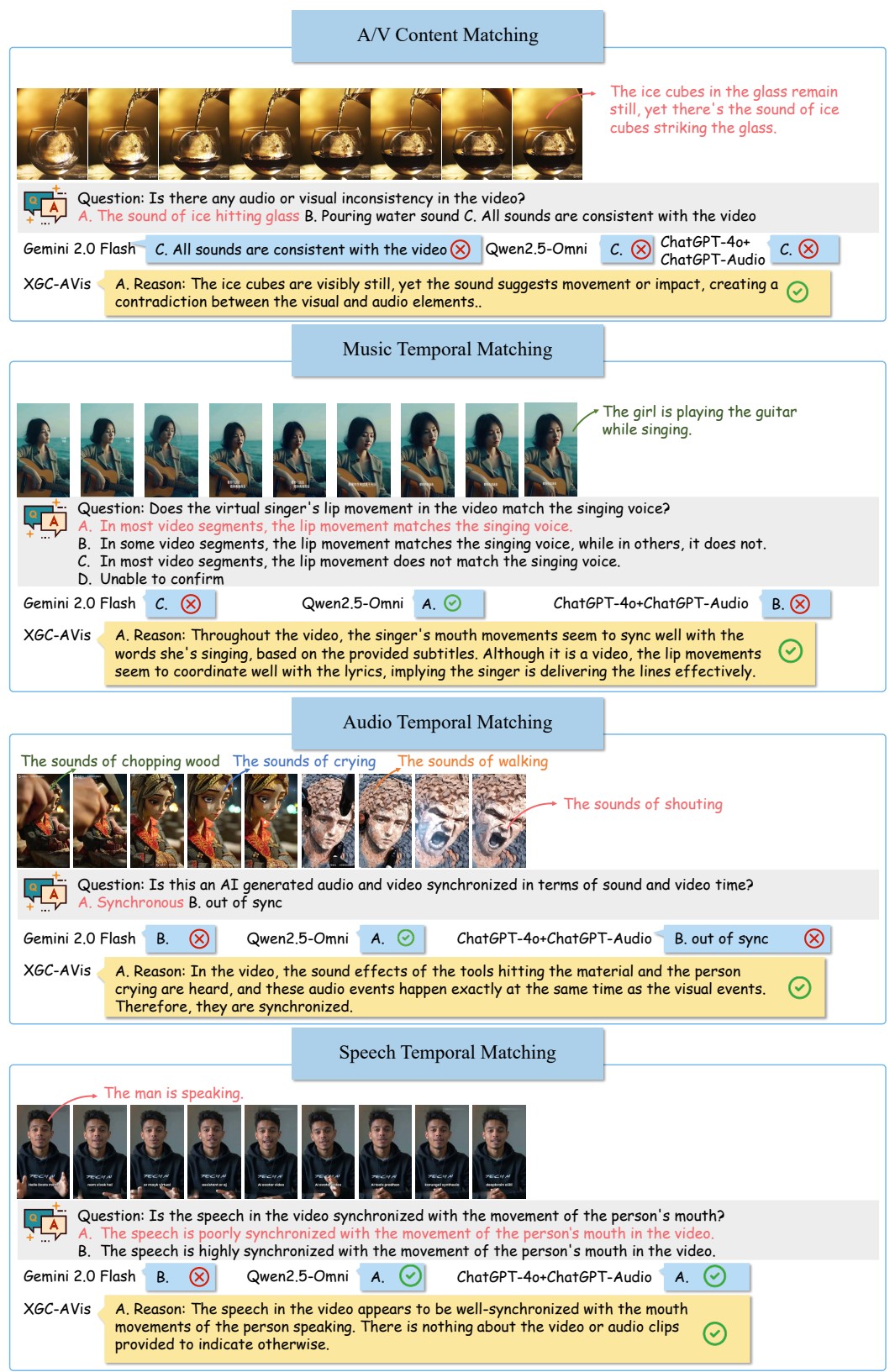

Figure 8: Examples of MLLM and XGC-AVis responses in the A/V quality perception category, including A/V content matching, music temporal matching, audio temporal matching and speech temporal matching.

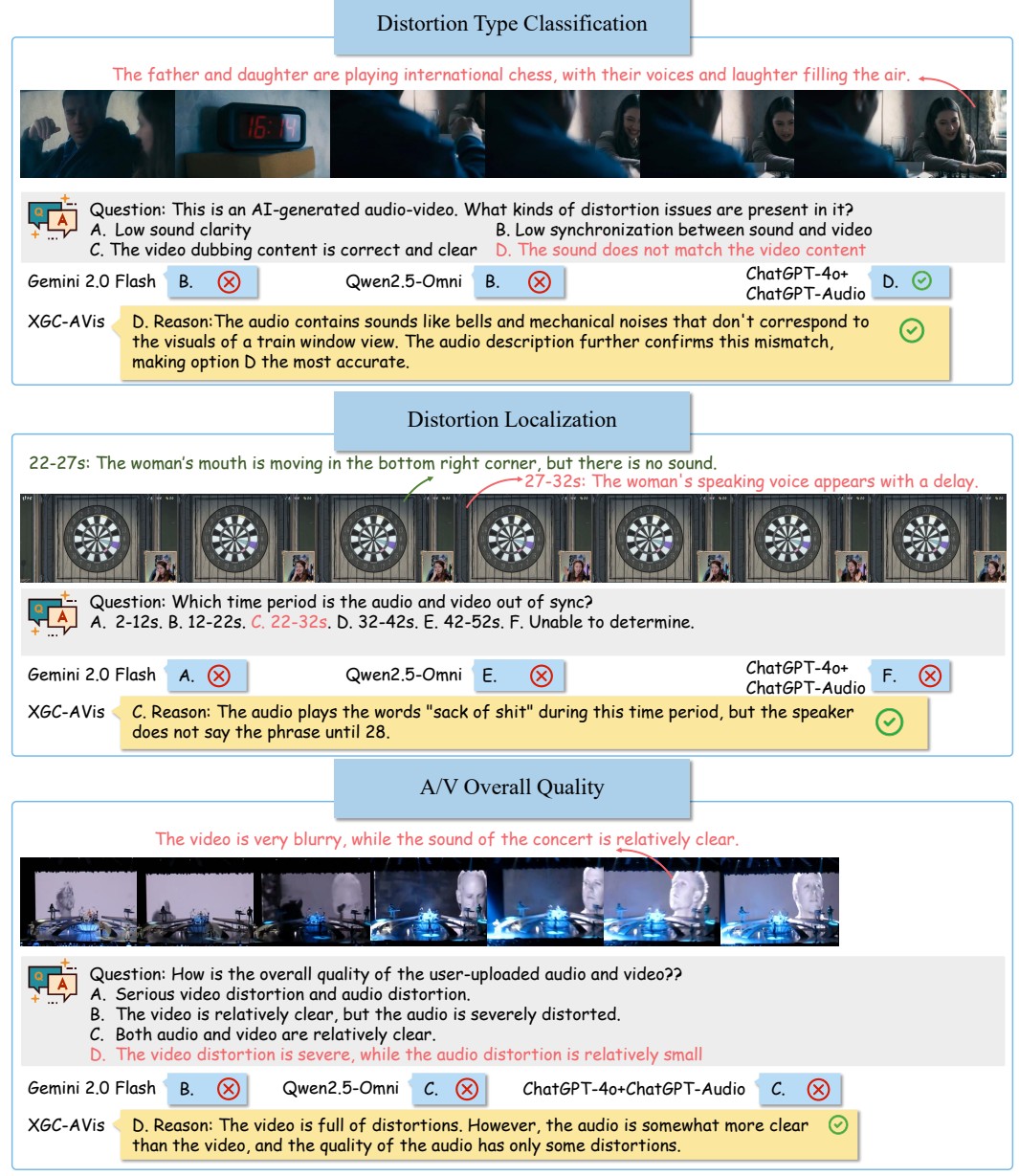

Figure 9: Examples of MLLM and XGC-AVis responses in the A/V quality perception category, including distortion type classification, distortion localization and A/V overall quality.

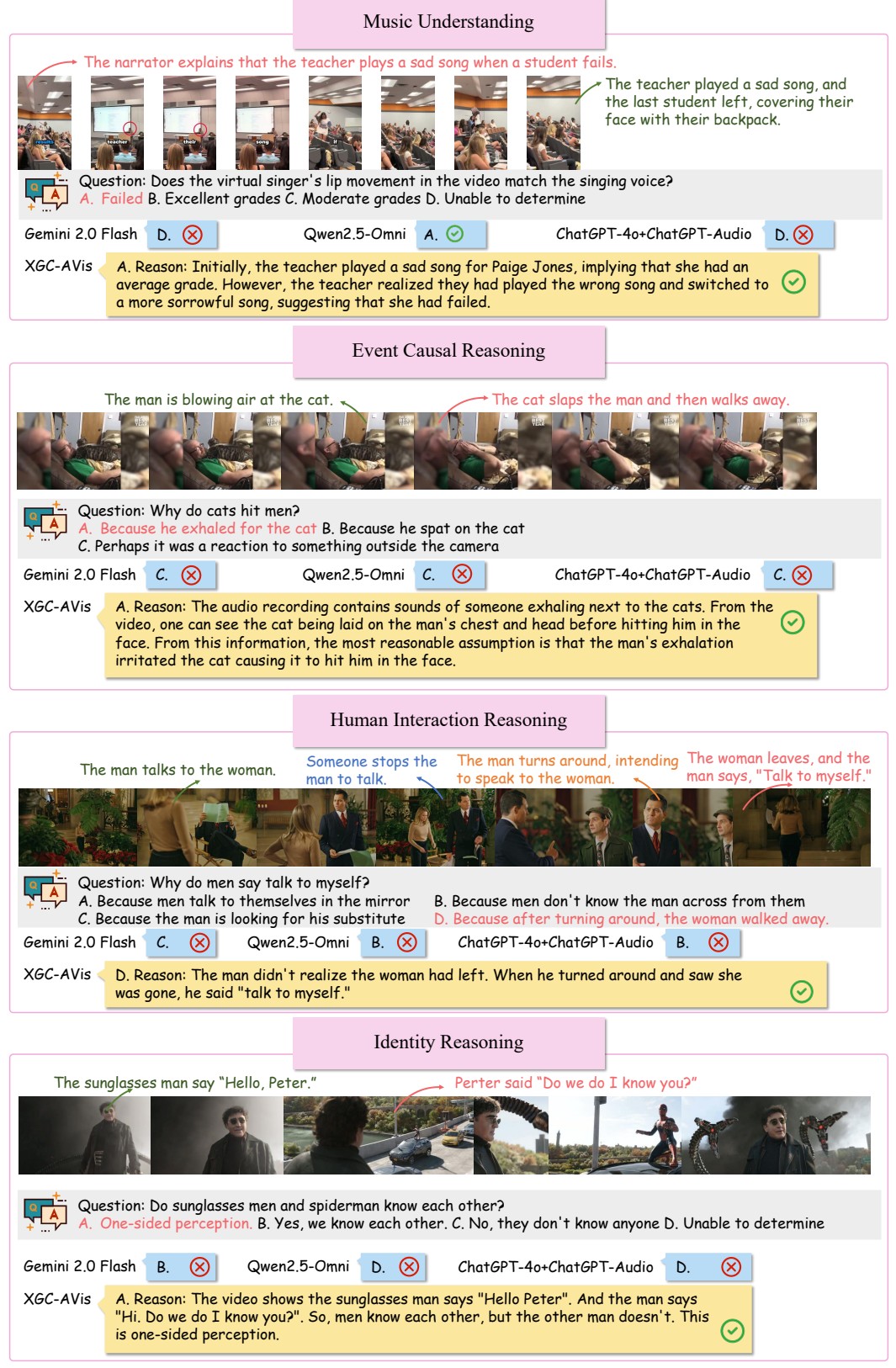

Figure 10: Examples of MLLM and XGC-AVis responses in the A/V reasoning category, including music understanding, event causal reasoning, human interaction reasoning and identity reasoning.

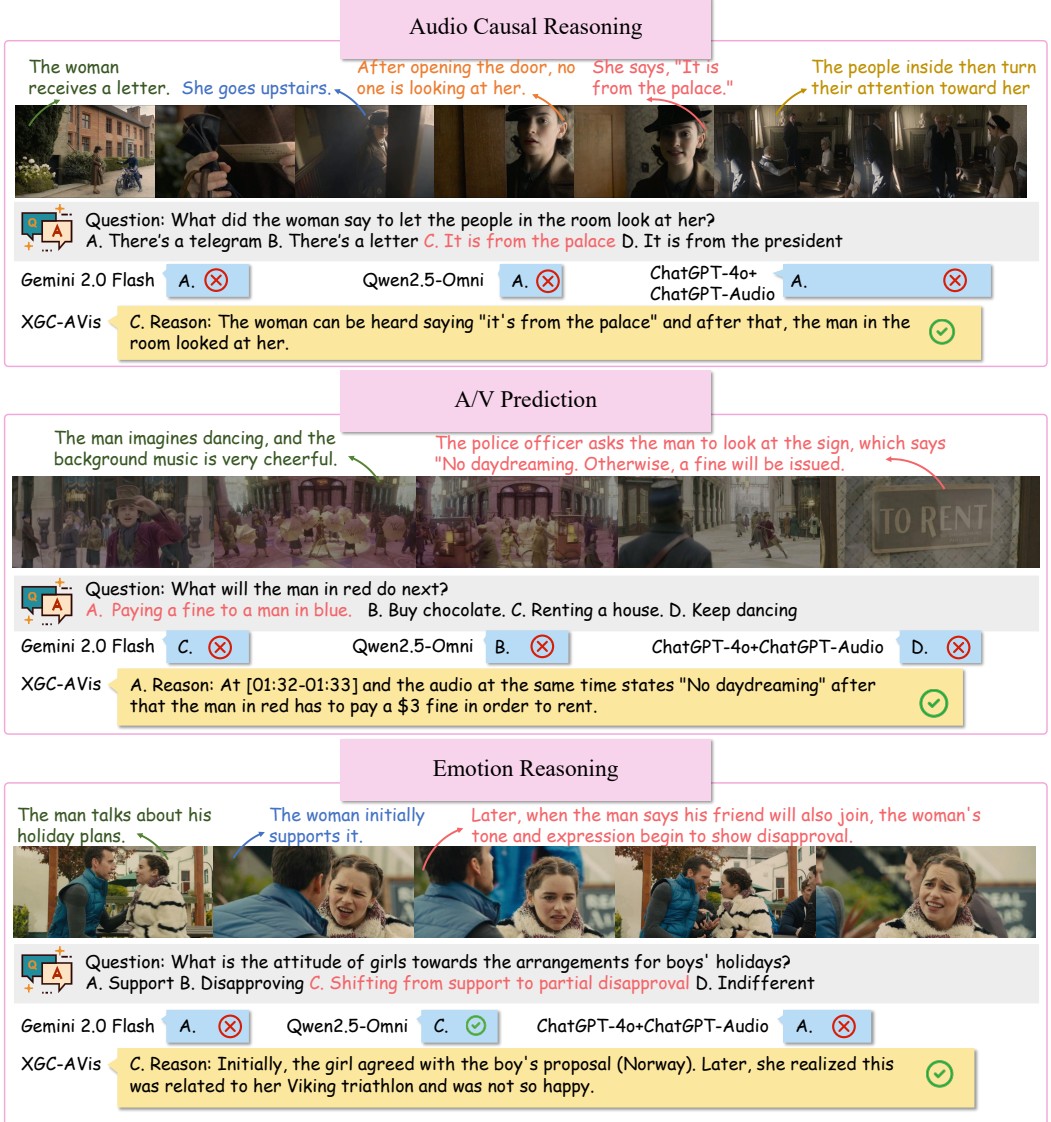

Figure 11: Examples of MLLM and XGC-AVis responses in the A/V reasoning category, including audio causal reasoning, A/V prediction, and emotion reasoning.

