# OpenReview forum: "XGC-AVis: Towards Audio-Visual Content Understanding with a Multi-Agent Collaborative System"
_ICLR.cc/2026/Conference — Submitted to ICLR 2026_

### Official Review · Reviewer_yfqm · 2025-11-01

**Soundness:** 3
**Presentation:** 3
**Contribution:** 3
**Rating:** 6
**Confidence:** 3

**Summary:**

XGC-AVis introduces a training-free multi-agent pipeline: (1) Perception, (2) Planning, (3) Execution and (4)Reflection, which enhance temporal alignment and quality perception in MLLMs. Inside XFC-AVis, two planners identify time segments relevant to a question, two executors reason over them, and a decider resolves conflicts, producing final answers.

The paper also presents XGC-AVQuiz, a new benchmark with 2,232 videos and 2,685 QA pairs spanning PGC, UGC, and AIGC sources across four categories (recognition, localization, reasoning, and quality perception). Experiments show consistent gains over both open- and closed-source MLLMs. Ablation results confirm that the dual-planner and decider design is key to improvements.

**Strengths:**

- Writing is logical and easy to follow, motivation is clear
- The paper introduces a novel multi-agent pipeline (Perception–Planning–Execution–Reflection) that improves A/V reasoning without extra training.
- Evaluations cover multiple open- and closed-source MLLMs, two benchmarks, and detailed ablations.

**Weaknesses:**

- The first weakness is the multi-agent framework increases computational cost as slove one question, needs more llms/models envloves in. The author should quantifying latency or cost trade-offs.
- I appreciate the author propose a new benchmark, and do a lot testing on this benchmark. However, some tasks in XGC-AVQuiz overlap conceptually with existing A/V QA datasets (e.g., AVQA, Daily-Omni), can the author also report on those datasets, and compare with traditional models?
- For the model: while the multi-agent design is novel, it mainly combines existing reasoning and planning components rather than introducing a fundamentally new learning mechanism or model architecture.

**Questions:**

Please see weakness part.

---

> ### Author Response · Authors · 2025-12-03
>
> We would like to thank the reviewer for the constructive and valuable feedback. We have addressed your concerns point-by-point below.
>
> **[W1] Computational Cost**
>
> We evaluated the average inference time across varying video lengths (10s, 30s, 1min, 5min) on two NVIDIA RTX 6000 (96GB). For the baseline, Gemini 2.0 Flash uses a fixed 15-frame input strategy, resulting in constant inference time. For XGC-AVis, we report the full pipeline execution time (assuming the "worst-case" scenario where the two executors are distinct).
>
> Method|10s|30s|1min|5min
> :-|:-:|:-:|:-:|:-:
> Gemini 2.0 Flash | 21.76s | 22.72s | 22.59s | 23.57s
> XGC-AVis (Ours) | 60.14s | 62.21s | 64.21s | 65.14s
>
> While XGC-AVis has higher latency due to its multi-step architecture, we highlight two key points:
>
> - **Stability:** Inference time remains highly stable as video length increases. Processing a 5-minute video takes only ~5s longer than a 10-second one, demonstrating efficient scaling without exponential cost.
>
> - **Trade-off:** We prioritize performance over speed. The extra time is a necessary trade-off to enable fine-grained cross-modal reasoning that monolithic MLLMs lack.
>
>
> **[W2] Generalization on External Benchmarks**
>
> To demonstrate the generalizability of XGC-AVis, we conducted extensive evaluations on WorldSense and Daily-Omni, comparing it against both traditional models and SOTA baselines.
>
> - **Performance on WorldSense:** On WorldSense, XGC-AVis (51.1%) consistently outperforms the baseline Gemini 2.0 Flash (49.9%), demonstrating robust generalization capabilities across diverse categories
>
> WorldSense|Tech & Science|Culture & Politics|Daily Life|Film & TV|Performance|Games|Sports|Music|Average
> :-|:-:|:-:|:-:|:-:|:-:|:-:|:-:|:-:|:-:
> Qwen2-VL|33.5|29.0|28.4|33.6|30.3|32.3|34.7|38.5|32.4
> Qwen 2.5 Omni|47.8|49.8|43.6|43.8|48.3|39.1|43.5|47.3|45.4
> Gemini 2.0 Flash|54.1|50.1|50.4|49.0|52.3|48.5|45.9|47.9|49.9
> XGC-AVis (Ours)|**54.1**|**50.2**|**51.7**|**50.1**|**54.7**|**50.2**|**47.2**|**50.5**|**51.1**
>
> - **Performance on Daily-Omni:** We further compared XGC-AVis against state-of-the-art models on the Daily-Omni benchmark. XGC-AVis achieves a state-of-the-art average score of 71.5%, significantly surpassing GPT-4o (56.5%), the Daily-Omni baseline (61.8%), and Gemini 2.0 Flash (67.8%).
>
> Daily-Omni|A/V Event Alignment|Comparative|Context Understanding|Event Sequence|Inference Reasoning|30 s Subset|60s Subset|Avg
> :-|:-:|:-:|:-:|:-:|:-:|:-:|:-:|:-:
> Unified-IO-2XXL|44.1|51.2|38.9|40.5|57.8|61.7|46.7|48.4|47.5
> VideoLLaMA2|35.7|35.9|35.8|31.7|40.9|34.3|38.0|31.8|35.2
> Qwen2.5-Omni|25.6|31.3|26.4|25.8|35.1|29.7|26.7|30.0|28.2
> ChatGPT-4o|47.9|62.6|52.3|52.6|66.2|66.3|55.6|57.5|56.5
> Gemini 2.0 Flash|Lite|55.0|64.9|58.0|54.3|74.0|72.0|62.4|60.0|61.3
> Gemini 2.0 Flash|62.2|73.3|63.7|63.7|76.6|75.4|67.2|68.6|67.8
> Daily-Omni|51.7|68.7|60.1|59.3|78.6|71.4|64.0|59.3|61.8
> XGC-AVis (Ours)|**63.5**|**77.1**|**68.4**|**64.4**|**85.1**|**82.3**|**71.6**|**71.5**|**71.5**
>
> **[W3] Clarification on Novelty**
>
> We respectfully clarify that the novelty and contribution of our multi-agent design XGC-AVis lie in two key aspects:
>
> - **Flexible Framework:** While XGC-AVis does not rely on fine-grained training, this is a deliberate design choice. It functions as a model-agnostic framework that can be flexibly integrated with various Audio-Video MLLMs to universally enhance their capabilities.
>
> - **Training-Free Performance Boost:** XGC-AVis achieves substantial performance improvements without requiring additional training resources. This highlights a novel, resource-efficient paradigm for boosting model understanding without the computational burden of retraining.

---

### Official Review · Reviewer_e7WQ · 2025-11-01

**Soundness:** 2
**Presentation:** 2
**Contribution:** 2
**Rating:** 4
**Confidence:** 4

**Summary:**

This paper proposes XGC-AVis, a four-stage (perception, planning, execution, reflection) multi-agent framework designed to enhance audio-visual temporal alignment and key segment retrieval for multimodal large language models (MLLMs). It also introduces XGC-AVQuiz, a benchmark with 2,232 videos (PGC/UGC/AIGC) and 2,685 QA pairs covering 20 tasks, including a novel "quality perception" dimension. Experimental results claim XGC-AVis outperforms existing MLLMs on XGC-AVQuiz and Daily-Omni.

**Strengths:**

- The XGC-AVQuiz benchmark attempts to address the limitation of existing datasets by integrating PGC, UGC, and AIGC scenarios, and introduces a "quality perception" task to evaluate MLLMs’ ability to assess audio-visual synchronization and coherence—an underexplored area in current benchmarks.

- The four-stage multi-agent design of XGC-AVis provides a approach to tackle audio-visual temporal alignment, which is a known challenge for MLLMs.

**Weaknesses:**

- The paper is lack of novelty. It claims to enhance audio-video understanding of MLLMs via XGC-AVis, but it fails to address the fundamental issue of poor fine-grained capabilities in audio-video MLLMs. XGC-AVis only adopts a "component-stitching" approach (e.g., using Deepgram for subtitles and r1-aqa for audio descriptions, Section 3) to interweave unimodal outputs, without realizing feature-level or semantic-level deep cross-modal interaction—leaving the core problem of weak fine-grained cross-modal association unresolved
- The paper explicitly states that XGC-AVis "interweaves video frames, audio segments, subtitles, and audio descriptions to form coherent multimodal units" (Section 3, 1-26), but it completely ignores the feature space gap between audio and visual modalities—an issue that directly undermines the "coherence" of the resulting multimodal units.
- Directly concatenating or interleaving unaligned features forces the subsequent planners/executors  to handle inconsistent feature spaces, which can lead to two critical issues: The model may over-rely on one modality (e.g., visual features, which are more structured) and ignore audio features, weakening the intended "multimodal collaboration". The paper does not acknowledge or address these risks.
- XGC-AVis integrates multiple expert models/tools (Deepgram, r1-aqa, Aria, Qwen2.5-Omni, Gemini 2.0 Flash) to form a multi-expert ensemble system, while the comparison baselines in Table 2 are standard single MLLMs . This comparison is unfair, as XGC-AVis leverages "collective intelligence" and higher computational costs that standard MLLMs lack, yet the paper only emphasizes performance advantages without acknowledging such trade-offs.
- The paper uses Aria (Planner 1) and Qwen2.5-Omni (Planner 2), but Aria "can only process video and text" while Qwen2.5-Omni "processes video, audio, and text" (Appendix A.3). Why combine a unimodal (video-text) planner with a multimodal (video-audio-text) planner? Does this combination introduce redundancy, and if not, what unique value does each planner provide?
- Tables 2 only report "average accuracy" for broad tasks (e.g., A/V Localization) but not task-specific metrics for time alignment—such as Temporal Localization Accuracy (TLA): The percentage of cases where the model’s predicted time segment overlaps with the ground-truth by ≥50% IoU
- The "perception" stage uses both Deepgram (speech-to-subtitle) and r1-aqa (audio descriptor). However, the paper does not clarify whether their outputs are redundant—for example, if Deepgram’s subtitles already capture speech content, what unique information does r1-aqa’s audio description add (e.g., background noise, emotion)?
- Generally, audio and video in datasets are consistent (Section 4.1), so audio should only bring marginal improvements for most tasks. However, Table 2 shows a significant performance boost when adding audio (e.g., Qwen2.5-Omni’s A/V Recognition accuracy jumps ~16% from Vid. to Vid.+Aud.). This is unusual. Additionally, if video-only inputs already achieve high accuracy on supposed "audio-driven" tasks, it implies potential flaws in the evaluation’s task design, undermining its credibility.
- The paper does not clarify whether the questions in Table 2 differ between the "video-only" and "video+audio" settings. Without disclosing the distribution of modality-dependent questions, the observed performance boost from adding audio may stem from question design bias (forcing audio reliance) rather than true multimodal understanding, making the results uninterpretable.
- The authors do not evaluate whether questions can be answered with a single modality (e.g., audio-only). The experimental input settings (Section 5.1) only include video-only, video+audio, and video+audio+subtitle, without testing audio-only inputs. This omission means it is impossible to verify if multimodal integration is truly necessary—for example, some A/V reasoning tasks may be solvable via video alone (facial expressions) or audio alone (tone), rendering XGC-AVis’s multimodal design redundant.

**Questions:**

See weakness

---

> ### Author Response · Authors · 2025-12-03
>
> We would like to thank the reviewer for the constructive and valuable feedback. We have addressed your concerns point-by-point below.
>
> **[W1] Clarification on Novelty**
> We respectfully clarify that the novelty and contribution of our work lie in three key aspects:
>
> - **Comprehensive Benchmark Construction:** We introduce a significantly richer audio-video benchmark that expands upon AIGC content and incorporates diverse evaluation dimensions, specifically targeting perceptual capabilities. This fills a gap in existing multi-modal evaluation.
>
> - **Flexible Framework:** While XGC-AVis does not rely on fine-grained training, this is a deliberate design choice. It functions as a model-agnostic framework that can be flexibly integrated with various Audio-Video MLLMs to universally enhance their capabilities.
>
> - **Training-Free Performance Boost:** XGC-AVis achieves substantial performance improvements without requiring additional training resources. This highlights a novel, resource-efficient paradigm for boosting model understanding without the computational burden of retraining.
>
> **[W2、W3] Feasibility of Interleaved Modality Integration**
>
> We disagree with the notion that our "interleaving" approach undermines coherence due to feature space gaps. We argue that semantic-level alignment via MLLMs is a proven and effective alternative to rigid feature-level fusion, supported by both existing literature and our empirical results.
>
> - **Literature Support:** Recent advancements in MLLMs have demonstrated that interleaving visual tokens with textual descriptions (or timestamps) is a highly effective strategy for bridging the semantic gap. Aria (Rhymes AI, 2024) enhances temporal understanding by explicitly interleaving timestamp text with video frames, proving that text-visual interleaving effectively aids the model in grounding temporal information. Flamingo (DeepMind, 2022), VILA (NVIDIA, 2024) and MM1 (Apple, 2024) demonstrate that training on interleaved image-text data enables models to perform complex reasoning across modalities. These works highlight that the architecture of interleaving data inputs is crucial for unlocking few-shot learning and in-context reasoning capabilities in MLLMs.
>
> - **Ablation Study:**  We conduct an ablation study on the interleaving step to quantify its contribution. As shown in the table below, removing the interleaved format ("*w/o* interleave") leads to a significant performance drop. This confirms that the "interweaving" mechanism effectively enhances fine-grained cross-modal reasoning capabilities.
>
> Methods|A/V Recognition|A/V Localization|A/V Perception|A/V Reasoning|Average
> :-|:-:|:-:|:-:|:-:|:-
> XGC-AVis *w/o* interleave|57.2|47.1|45.1|67.5|54.2
> XGC-AVis (Ours)|59.5|51.7|51.0|69.3|58.2
>
>
> **[W4] Computational Cost**
>
> We evaluated the average inference time across varying video lengths (10s, 30s, 1min, 5min) on two NVIDIA RTX 6000 (96GB). For the baseline, Gemini 2.0 Flash uses a fixed 15-frame input strategy, resulting in constant inference time. For XGC-AVis, we report the full pipeline execution time (assuming the "worst-case" scenario where the two executors are distinct).
>
> Method|10s|30s|1min|5min
> :-|:-:|:-:|:-:|:-:
> Gemini 2.0 Flash | 21.76s | 22.72s | 22.59s | 23.57s
> XGC-AVis (Ours) | 60.14s | 62.21s | 64.21s | 65.14s
>
> While XGC-AVis has higher latency due to its multi-step architecture, we highlight two key points:
>
> - **Stability:** Inference time remains highly stable as video length increases. Processing a 5-minute video takes only ~5s longer than a 10-second one, demonstrating efficient scaling without exponential cost.
>
> - **Trade-off:** We prioritize performance over speed. The extra time is a necessary trade-off to enable fine-grained cross-modal reasoning that monolithic MLLMs lack.
>
> **[W5] Rationale for Planner Selection**
>
> We selected Aria and Qwen2.5-Omni to leverage their complementary strengths rather than creating redundancy:
>
> - **Qwen2.5-Omni (Native Audio Perception):** It was the SOTA open-source audio-video MLLM at the time, offering superior performance in sound understanding and quality perception, where other models often fall short. It provides the crucial raw audio processing capability.
>
> - **Aria (Temporal Grounding):** Aria integrates timestamps with video frames. This makes it suited to process our interleaved multimodal units, providing the precise temporal reasoning and structure that complements Qwen2.5-Omni.

---

> ### Author Response · Authors · 2025-12-03
>
> **[W6] Evaluation on Temporal Localization Accuracy (TLA)**
>
> Our metric in Table 2 reports accuracy because most A/V Localization tasks in our benchmark focus on cross-modal content association (e.g., describing visual details corresponding to a specific audio event).
>
> However, to address the need for precise time-alignment metrics, we filtered the subset of questions specifically requiring numerical timestamp answers to calculate TLA. The results are as follows:
>
> Model|Qwen 2.5 Omni|Gemini 2.0 Flash|XGC-AVis
> :-|:-:|:-:|:-
> TLA|24.1%|26.4%|**28.6%**
>
> XGC-AVis achieves the highest TLA (28.6%), outperforming the strong closed-source baseline Gemini 2.0 Flash. This further validates that our interleaved architecture significantly enhances fine-grained temporal grounding.
>
> **[W7] Distinct Roles of Deepgram and r1-aqa**
>
> Deepgram and r1-aqa are not redundant; they serve complementary roles to ensure comprehensive audio perception:
>
> - **Deepgram**: In our experiments, r1-aqa exhibited limited robustness in speech recognition, often producing hallucinatory translations, e.g., misinterpreting English speech as Chinese. **Therefore, we employ Deepgram strictly for accurate Automatic Speech Recognition (ASR) to generate subtitles.**
>
> - **r1-aqa**: We feed the audio along with Deepgram's subtitles into r1-aqa. By providing the speech text, we guide r1-aqa to focus on non-speech acoustic details, such as sound effects, background music, and emotional tone. The specific prompt templates used for r1-aqa are as follows:
>
>     **Prompt Templates for Audio Descriptor with Speech:**
>
>     *User: The subtitle of this audio: [subtitle]. Please describe the background sounds and music. Do not respond using list or dictionary formats. Instead, write your response in full sentences.*
>
>     **Prompt Templates for Audio Descriptor without Clear Speech:**
>
>     *This audio does not contain clear speech. Please the background sounds, emotion and music. Do not respond using list or dictionary formats. Instead, write your response in full sentences.*
>
> This pipeline leverages Deepgram's precision for text and r1-aqa's semantic understanding for environmental and emotional audio cues.
>
> **[W8] Reason for Performance Gain from Audio Integration**
>
> During data collection, annotators were required to provide separate answers based on video-only input and audio-video input. We only included samples where the video-only answer differed from the audio-video input answer. This filtering guarantees that visual information alone is insufficient to solve the task, thereby naturally resulting in the substantial performance boost seen when the audio modality is correctly integrated.
>
> **[W9] Consistency of Questions Across Modalities**
>
> We clarify that the questions in Table 2 are identical for both the "video-only" and "video+audio" settings. The experimental setup holds the questions constant and varies only the input modalities provided to the model. This ensures that the observed performance boost is solely due to the model's ability to leverage the audio information, rather than any variation in question design.
>
> **[W10] Evaluation of Audio-Only Performance**
>
> To address the concern that tasks might be solvable via a single modality, we extended our evaluation to include an Audio-only setting. We compared performance across three input configurations: Video-only (Vid.), Audio-only (Aud.), and Video+Audio (V+A).
>
> Methods||A/V Recognition|||A/V Localization|||A/V Perception|||A/V Reasoning|||Average| |
> :-|:-:|:-:|:-:|:-:|:-:|:-:|:-:|:-:|:-:|:-:|:-:|:-:|:-:|:-:|:-:
> Modality|Vid.|Aud.|V+A|Vid.|+Aud.|V+A|Vid.|+Aud.|V+A|Vid.|+Aud.|V+A|Vid.|+Aud.|V+A
> PandaGPT|32.4%|37.8%|35.8%|31.3%|30.9%|32.3%|39.6%|31.6%|29.4%|34.9%|37.0%|35.5%|36.4%|34.0%|32.5%
> GroundingGPT|38.9%|38.2%|40.9%|35.8%|28.8%|35.8%|30.1%|29.1%|29.5%|37.6%|32.5%|48.8%|34.2%|31.2%|38.0%
> Bubogpt|23.0%|27.7%|13.5%|15.6%|26.0%|17.7%|19.4%|34.2%|21.2%|15.1%|25.0%|15.3%|17.9%|29.5%|18.0%
> Unified-IO-2|30.4%|31.8%|30.1%|29.5%|21.2%|29.9%|38.3%|39.6%|34.9%|28.1%|29.2%|26.4%|33.0%|33.3%|30.9%
> Unified-IO-2 XL|34.1%|35.8%|35.1%|24.7%|26.7%|25.3%|28.0%|28.2%|27.5%|27.4%|27.4%|27.4%|28.1%|28.6%|28.1%
> Unified-IO-2 XXL|32.4%|42.6%|33.4%|27.1%|30.9%|29.5%|38.7%|40.5%|37.3%|30.4%|33.0%|31.1%|34.0%|37.2%|33.9%
> Qwen2.5-Omni|47.6%|52.4%|53.7%|40.6%|39.6%|33.0%|37.1%|44.8%|41.8%|45.0%|49.3%|55.1%|41.3%|46.6%|46.7%
> Gemini 2.0 Flash|45.9%|51.4%|55.7%|43.4%|39.2%|49.3%|34.3%|36.0%|38.1%|48.5%|60.6%|68.1%|41.3%|46.4%|51.4%
>
> The results indicate that XGC-AVQuiz cannot be effectively solved through a single modality. **The performance gap demonstrates that neither visual nor audio cues alone are sufficient.** Crucially, for the strongest models (Gemini 2.0 Flash and Qwen2.5-Omni), the combined V+A input consistently yields the highest average performance. This confirms the validity and necessity of XGC-AVis's multimodal design.

---

### Official Review · Reviewer_9Q2y · 2025-11-01

**Soundness:** 2
**Presentation:** 3
**Contribution:** 2
**Rating:** 4
**Confidence:** 4

**Summary:**

This paper introduces **XGC-AVis**, a multi-agent framework designed to enhance multimodal large language models’ (MLLMs) ability to align audio and video temporally and to improve the efficiency of retrieving key video segments. The system operates through four stages—perception, planning, execution, and reflection—enabling coordinated agent interaction for more robust multimodal reasoning.

To evaluate such capabilities, the authors also propose **XGC-AVQuiz**, a new benchmark that comprehensively assesses MLLMs’ understanding **across real-world and AI-generated scenarios**. XGC-AVQuiz features both professionally generated (PGC), user-generated (UGC), and AI-generated (AIGC) videos spanning multiple domains, and it introduces a quality perception dimension that tests synchronization, coherence, and audio-visual quality awareness. Experimental results suggest that existing MLLMs struggle in these aspects, while XGC-AVis achieves measurable improvements without additional training.

**Strengths:**

**High-quality benchmark contribution.**
XGC-AVQuiz provides a more comprehensive, multi-level evaluation benchmark for multimodal reasoning, covering diverse data sources (PGC, UGC, AIGC) and introducing the novel dimension of quality perception. This significantly enriches the evaluation landscape for audio–visual MLLMs.

**Exploratory insights into multi-agent design.**
XGC-AVis presents valuable design insights for constructing multi-agent collaborative systems, offering a practical exploration into how workflow-level designs can help MLLMs overcome model-level limitations when dealing with complex multimodal understanding tasks.

**Weaknesses:**

- **Lack of empirical support for efficiency claims.**
  The paper states that **XGC-AVis** *"improves the efficiency of retrieving key video segments”*, but presents no quantitative evidence or analysis to support this. Without metrics or user-study results, this claim remains unsubstantiated.

- **Insufficient citations and context in the introduction.**
  The discussion of MLLMs and representative models cites mainly vision–language (e.g., Llava-OneVision, InternVL3) or pure language models , but fails to reference existing multimodal reasoning efforts that go beyond surface-level fusion [1–3]. This weakens the argumentative foundation of the introduction.

- **Unconvincing “data source bias” argument.**
  The claim that existing datasets are dominated by user-generated content (UGC) and lack professionally generated content (PGC) is poorly justified. Given that platforms like YouTube already contain high-quality professional material, the UGC–PGC distinction should be better defined or empirically supported.

- **Loose connection between Related Work and the proposed system.**
  The Related Work section focuses mainly on architectural summaries of MLLMs, while XGC-AVis is a workflow-level, multi-agent system. To strengthen positioning, the authors should include relevant **agent-level** or **system-level** works such as [4–5].

- **Ambiguous baseline description.**
  It is unclear whether *VideoLLaMA2* or *VideoLLaMA2.1-AV* is used as the baseline. These two differ substantially in multimodal input handling, and the choice directly affects the validity of the comparison.

- **Limited comparative and generalization experiments.**
  (1) Table 1 compares only VLM and Qwen2.5-Omni; adding *Qwen2.5-VL* would provide a fairer multimodal baseline.
  (2) Reporting XGC-AVis’s performance on an external benchmark such as *WorldSense* would help demonstrate generalizability.

- **Missing ablations on key system components.**
  While the data ablations are informative, the paper lacks **component-level** analysis. For example, how does the *Interleave* step in “Align multimodal data” affect performance? Additionally, replacing *Gemini 2.0 Flash* (executors/decider) with open-source alternatives would help isolate whether gains stem from the multi-agent design or the strength of the base models.

---
# Reference

[1] Aligned Better, Listen Better for Audio-Visual Large Language Models, ICLR 2025, https://arxiv.org/abs/2504.02061

[2] HumanOmni: A Large Vision-Speech Language Model for Human-Centric Video Understanding, https://arxiv.org/abs/2501.15111

[3] SALMONN family: A suite of advanced multi-modal LLMs,https://github.com/bytedance/SALMONN

[4] ContextAgent: Context-Aware Proactive LLM Agents with Open-World Sensory Perceptions, https://arxiv.org/abs/2505.14668

[5] Daily-Omni: Towards Audio-Visual Reasoning with Temporal Alignment across Modalities

**Questions:**

The key concerns are outlined in the weaknesses section. Should the authors **provide convincing responses or improvements** addressing these points, **I would be open to increasing my evaluation score**.

---

> ### Author Response · Authors · 2025-12-03
>
> We would like to thank the reviewer for the constructive and valuable feedback. We have addressed your concerns point-by-point below.
>
> **[W1] Experimental Validation of Efficiency Claims**
>
> To substantiate our claim that XGC-AVis efficiently retrieves key video segments, we conducted two quantitative evaluations:
>
> - **Model Performance:** We replaced the key video segments selected by the planners in XGC-AVis with the entire video. As shown in the table below, processing the full video ("*w/o* key video segments") actually degrades performance across all categories.
>
> Method|A/V Recognition| A/V Localization|A/V Perception|A/V Reasoning
> :-|:-:|:-:|:-:|:-:
> XGC-Avis *w/o* key video segments|56.1|50.1|45.1|68.3
> XGC-AVis (Ours)|59.5|51.7|51.0|69.3
>
> - **Human Validation:** We conducted a user study with $10$ participants to verify if the key video segments contained sufficient information. Participants were asked to answer questions based only on the key segments selected by Planner 1 or Planner 2, without seeing the full video. The human accuracy rates were 87% (Planner 1 segments) and 84% (Planner 2 segments).
>
> These results demonstrate that XGC-AVis successfully extracts the most informative segments, achieving high data efficiency.
>
> **[W2] Introduction Supplement**
>
> We thank the reviewer for pointing out these crucial references. We will incorporate these citations into the Introduction of the final revision as follows:
>
> Therefore, a growing number of works have begun to explore deeper audio–visual understanding and alignment in large multimodal models $[1–3]$. For instance, HumanOmni $[2]$ integrates human speech with video for human-centric understanding, SALMONN $[3]$ advances the comprehension of general audio including music and background sounds, and Dolphin $[1]$ specifically targets the alignment of audio and visual modalities in the temporal dimension. These efforts highlight the critical importance of evaluating a model’s capacity for modeling non‑speech auditory information and fine-grained cross‑modal temporal alignment—such as A/V synchronization and temporal localization.
>
> **[W3] Comparison of Data Sources**
>
> To address the concern regarding the definition of PGC, we quantify the distinction between UGC and PGC through two quantitative metrics: video resolution (visual fidelity) and shot change frequency (editing complexity). We conducted a quantitative analysis comparing XGC-AVQuiz against WorldSense and Daily-Omni. Shot changes are detected using PySceneDetect based on pixel-level content discontinuities between consecutive frames.
>
> - **Resolution Distribution:** Existing datasets mainly contain low-resolution content, while XGC-AVQuiz introduces high-definition standards.
>
> Resolution|1080p|720p|360p
> :-|:-:|:-:|:-
> WorldSense|0|9|1653
> Daily-Omni|0|0|684
> XGC-AVQuiz (Ours)|609|945|1168
>
> - **Editing Dynamics:** Higher shot change frequency indicates professional editing and dynamic storytelling
>
> Resolution|WorldSense|Daily-Omni|XGC-AVQuiz (Ours)
> :-|:-:|:-:|:-
> Shot Change Freq|0.18|0.19|0.21
>
> The results reveals that while platforms like YouTube host diverse content, the specific datasets used in prior benchmarks (WorldSense, Daily-Omni) consist almost exclusively of 360p, static videos. In contrast, XGC-AVQuiz contains a substantial proportion of high-resolution (1080p/720p) content and exhibits significantly higher editing dynamics. This evidence validates our claim that XGC-AVQuiz fills a critical gap by providing professionally produced, high-fidelity data that prior benchmarks lack.
>
> **[W4] More Related Work**
>
> We thank the reviewers for the suggestion. Accordingly, we will add the following content in the related work:
>
> Recent research has started to explore agent‑level and system‑level paradigms that mirror complex real-world workflows. For example, ContextAgent proposes a context‑aware proactive LLM agent that integrates open-world sensory perception with historical context to dynamically invoke tools. Daily-Omni employs a multi-agent approach specifically for audio-visual reasoning, emphasizing temporal alignment across modalities. Building upon these system-level innovations, XGC-AVis proposes a specialized multi-agent workflow that enhances fine-grained understanding by explicitly interweaving video frames, audio segments, subtitles, and audio descriptions.
>
> **[W5] Clarification of Baseline**
>
> We clarify that all experiments reported in the paper utilized VideoLLaMA2.1-AV. We did not use VideoLLaMA2 without audio‑visual support.

---

> ### Author Response · Authors · 2025-12-03
>
> **[W6, W7] More Ablation Study**
>
> Thank you for the valuable suggestions. We have significantly expanded our experiments to address the concerns regarding baselines, generalization, and component contribution.
>
> - **Added a VLM Baseline:** Qwen2.5-VL achieves an average accuracy of 40.1%, which is significantly lower than Qwen2.5-Omni (45.4%).
>
> - **Experiments on WorldSense:** On WorldSense, XGC-AVis (51.1%) consistently outperforms the baseline Gemini 2.0 Flash (49.9%), demonstrating robust generalization capabilities across diverse categories
>
> - **Impact of Interleaving:** Removing the "Interleave" step causes a significant performance drop from 58.2% to 54.2%. This confirms that our interleaving strategy is critical for fine-grained cross-modal reasoning.
>
> - **Open-Source Feasibility:** We replaced Gemini 2.0-Flash with the open-source Qwen2.5-Omni for the executor/decider. The resulting system (50.8%) still outperforms the single Qwen2.5-Omni baseline (49.8%). This proves that the XGC-AVis architecture itself provides structural improvements, independent of the underlying model's strength.
>
> Methods||A/V Recognition|||A/V Localization|||A/V Perception|||A/V Reasoning|||Average| |
> :-|:-:|:-:|:-:|:-:|:-:|:-:|:-:|:-:|:-:|:-:|:-:|:-:|:-:|:-:|:-:
> Modality|Vid.|+Aud.|+Sub.|Vid.|+Aud.|+Sub.|Vid.|+Aud.|+Sub.|Vid.|+Aud.|+Sub.|Vid.|+Aud.|+Sub.
> Qwen2.5-VL|42.6|-|42.2|41.3|-|41.3|38.6|-|38.6|41.3|-|41.1|40.3|-|40.1
> Qwen2.5-Omni|47.6|55.4|53.7|40.6|37.5|33.0|37.1|45.1|41.8|45.0|57.9|55.1|41.3|49.8|46.7
>
>
> Model|Tech & Science|Culture & Politics|Daily Life|Film & TV|Performance|Games|Sports|Music|Average
> :-|:-:|:-:|:-:|:-:|:-:|:-:|:-:|:-:|:-:
> Qwen2-VL|33.5|29.0|28.4|33.6|30.3|32.3|34.7|38.5|32.4
> Qwen 2.5 Omni|47.8|49.8|43.6|43.8|48.3|39.1|43.5|47.3|45.4
> Gemini 2.0 Flash|54.1|50.1|50.4|49.0|52.3|48.5|45.9|47.9|49.9
> XGC-AVis (Ours)|**54.1**|**50.2**|**51.7**|**50.1**|**54.7**|**50.2**|**47.2**|**50.5**|**51.1**
>
> Methods|A/V Recognition|A/V Localization|A/V Perception|A/V Reasoning|Average
> :-|:-:|:-:|:-:|:-:|:-
> Qwen2.5 Omni|55.4|37.5|45.1|57.9|49.8
> XGC-AVis with Qwen2.5 Omni|55.7|38.1|47.1|58.1|50.8
> XGC-AVis *w/o* interval|57.2|47.1|45.1|67.5|54.2
> XGC-AVis (Ours)|**59.5**|**51.7**|**51.0**|**69.3**|**58.2**

---

### Official Review · Reviewer_bVyF · 2025-11-01

**Soundness:** 2
**Presentation:** 2
**Contribution:** 2
**Rating:** 4
**Confidence:** 4

**Summary:**

The submitted manuscript proposes XGC-AVis a multi-agent framework designed to enhance temporal alignment and audio-visual reasoning capabilities in MLLMs. The authors also introduce XGC-AVQuiz, a comprehensive benchmark comprising 2,232 videos covering both real-world and AI-generated content and 2,685 question–answer pairs, aimed at evaluating recognition, localization, quality perception, and reasoning abilities in audio-visual tasks. Extensive experiments verify the effectiveness of XGC-AVis, demonstrating a certain degree of novelty.

**Strengths:**

1. The manuscript clearly defines the problem and provides a well-motivated solution.
2. The construction of the XGC-AVQuiz benchmark is valuable and contributes to the development of the audio-visual research community.
3. The proposed XGC-AVis framework achieves superior performance across multiple benchmarks, and the ablation studies are comprehensive.
4. The paper is clearly written and easy to follow.

**Weaknesses:**

1. The related work section omits several directly relevant recent studies on multi-agent, multimodal, and video question-answering architectures and benchmarks, such as *MAGNET*, *VideoMultiAgents*, and *OmAgent*. The authors are encouraged to discuss distinctions from these works.
2. Given that MLLMs are used to generate distractors, how do the authors ensure that the benchmark does not inadvertently favor architectures used for generation or self-reflection?
3. Compared to single-agent or monolithic MLLM designs, how does XGC-AVis perform in terms of computational cost, especially for longer video inputs?
4. Since each query involves multiple *planner–executor* agents, the computational cost could be significant; a discussion or analysis of scalability is suggested.
5. Some experimental details require further clarification, for example, the input strategy for candidate time segments and the handling of inconsistent answers.
6. It would be helpful to elaborate on how conflicting segmentation results from different planners are weighted or fused during decision integration.

**Questions:**

My main questions are reflected in the Weaknesses Section.

---

> ### Author Response · Authors · 2025-12-03
>
> We would like to thank the reviewer for the constructive and valuable feedback. We have addressed your concerns point-by-point below.
>
> **[W1] More Related Works**
>
> We acknowledge these recent advancements and will incorporate them into our revision:
>
> - **MAGNET** focuses on retrieving segments across multiple videos. In contrast, XGC-AVis goes beyond retrieval to demand fine-grained recognition, localization, and quality perception within complex audio-video content.
>
> - **VideoMultiAgents** utilizes vision and text agents for QA. XGC-AVis distinguishes itself by integrating audio agents and emphasizing audio-video synchronization and deep cross-modal alignment.
>
> - **OmAgent** emphasizes long-video understanding via task decomposition. Conversely, XGC-AVis prioritizes the alignment of audio, video, and text across time and space, supported by a comprehensive benchmark designed specifically for these complex audio-visual scenarios.
>
> **[W2] Details of Distractors Generation**
>
> We clarify that our distractor generation strategy is adversarial rather than favorable to specific architectures. During dataset construction, we observed that many prior works manually design distractors, making it difficult to systematically control difficulty distribution. To address this, we prompt MLLMs with questions (without answers) and utilize their erroneous responses as high-quality distractors.
>
> This design serves two critical purposes:
>
> - It aligns the difficulty of the multiple-choice format with that of open-ended questions.
>
> - It prevents models from answering correctly simply via elimination. If a model's internal incorrect prediction appears as an option, the model is prone to selecting it, revealing its lack of genuine understanding.
>
> **[W3] Computational Cost**
>
> We evaluated the average inference time across varying video lengths (10s, 30s, 1min, 5min) on two NVIDIA RTX 6000 (96GB). For the baseline, Gemini 2.0 Flash uses a fixed 15-frame input strategy, resulting in constant inference time. For XGC-AVis, we report the full pipeline execution time (assuming the "worst-case" scenario where the two executors are distinct).
>
> Method|10s|30s|1min|5min
> :-|:-:|:-:|:-:|:-:
> Gemini 2.0 Flash | 21.76s | 22.72s | 22.59s | 23.57s
> XGC-AVis (Ours) | 60.14s | 62.21s | 64.21s | 65.14s
>
> While XGC-AVis has higher latency due to its multi-step architecture, we highlight two key points:
>
> - **Stability:** Inference time remains highly stable as video length increases. Processing a 5-minute video takes only ~5s longer than a 10-second one, demonstrating efficient scaling without exponential cost.
>
> - **Trade-off:** We prioritize performance over speed. The extra time is a necessary trade-off to enable fine-grained cross-modal reasoning that monolithic MLLMs lack.
>
> **[W4] Discussion of Scalability**
>
> We address the scalability concern by highlighting that XGC-AVis employs parallel execution via multiprocessing for both Plan Generation (Step 2) and Execution (Step 3). This design ensures that adding agents results in minimal latency overhead:
>
> - **Agent Scalability:** Adding more agents increases memory usage but has a minimal impact on inference time. For instance, increasing the number of planners from N=2 to N=4 doubled the GPU memory requirement (120GB $\rightarrow$ 240GB) but increased the average latency for a 1-minute video by only ~2 seconds (64.21s $\rightarrow$ 66.33s).
>
> - **Input Scalability:** To prevent runtime explosion with longer videos, we implement a maximum of 80 frames. This imposes an effective upper bound on runtime regardless of video duration, ensuring efficiency. (Please refer to Response 3 for specific inference time across different video lengths).

---

> ### Author Response · Authors · 2025-12-03
>
> **[W5,W6] Candidate Segment Processing and Decision Fusion**
>
> We clarify the handling of candidate time segments and  inconsistent answers below:
>
> In Step 1, we employ a uniform segmentation strategy, dividing the video into segments of at least 5 seconds, with a maximum of 80 segments. For each segment, corresponding subtitles and audio descriptions are extracted. In Step 2, each planner outputs a JSON object specifying candidate time segments and reasoning (see example below). If a planner fails to identify a specific segment, the entire video is used as a fallback.
> ```python
> {
>     "time_segments": [
>         {
>             "start_time": "01:16",
>             "end_time": "01:29",
>             "reasoning": "The man in front is seen riding past trees in the forested area. The specific count of trees he passes is not clearly visible, but the frames from 01:16 to 01:29 show multiple trees in the background as he navigates the trail."
>         }
>     ]
> }
> ```
> Planner outputs are mapped to the minimal set of uniform segments covering the proposed time range. These segments are then fed to the executors. If the executors produce conflicting answers, **we first take the union of the time segments identified by both planners to ensure maximum context coverage. Then we pass this combined context, along with both executors' initial answers and reasoning, to the decider.** The decider adjudicates the conflict to produce the final answer.
>
> In our experiments, conflicting answers occurred in approximately $27$% of cases. The decider successfully resolved these conflicts, yielding a $5.2$% accuracy improvement, which validates the effectiveness of this fusion strategy.

---

### Meta-Review · Area_Chair_jmne · 2026-01-07

**Summary:**

This paper proposes XGC-AVis, a training-free multi-agent framework for improving audio–visual temporal alignment and quality perception in multimodal large language models (MLLMs), along with XGC-AVQuiz, a new benchmark that includes PGC/UGC/AIGC content and introduces a novel quality perception dimension. Reviewers found the benchmark construction and the system-level exploration of multi-agent workflows to be interesting and potentially useful. However, the paper was consistently viewed as borderline, with multiple reviewers expressing concerns regarding the technical contribution, fairness of comparisons, computational cost, missing related works, and evaluation rigor. These concerns led to a majority of scores falling below the acceptance threshold.

**Reviewer Concerns:**

***Concerns substantially addressed by the rebuttal***

Benchmark positioning and PGC/UGC justification:
The authors added quantitative analyses, including resolution distributions and shot-change frequency statistics, to better support the claim that XGC-AVQuiz fills a gap in existing benchmarks by providing higher-fidelity and more professionally edited content.

Baselines and generalization:
Additional baselines (e.g., Qwen2.5-VL) and evaluations on external benchmarks (e.g., WorldSense and Daily-Omni) were included, strengthening the evidence for generalization beyond the proposed benchmark.

Ablations and modality controls:
The rebuttal introduced component ablations (e.g., removing the interleaving step), audio-only evaluations, and a temporal localization accuracy (TLA) metric, substantially improving the rigor of the empirical analysis.

***Outstanding / partially addressed concerns***

Nature of technical novelty:
Several reviewers may remain unconvinced that the contribution extends beyond a workflow-level orchestration of existing tools and models. While the system is carefully engineered, it does not introduce a new learning paradigm, representation, or modeling principle, and this limitation persists despite the rebuttal.

Computational cost and scalability:
The authors added detailed latency measurements across different video lengths and discussed scaling behavior under a capped-frame design. However, the reported inference time remains significantly slower than that of monolithic models. Moreover, the evaluation relies on sampling a fixed number of frames; for very long videos (e.g., hour-long inputs), a substantially larger number of frames would likely be required, further exacerbating the computational cost and scalability concerns.

Efficiency of key-segment retrieval:
While additional quantitative comparisons (key segments vs. full video) and human validation studies were provided, a more rigorous evaluation of key-segment retrieval accuracy (e.g., against ground-truth annotations) would further strengthen this claim.

Fairness of comparisons:
Despite the added cost analysis, comparisons between a multi-agent, multi-tool system and single-model baselines remain conceptually asymmetric. The observed performance gains may partly reflect increased system complexity rather than inherent improvements in model capability. This is a fundamental evaluation issue rather than something that can be resolved by adding more experiments.

Evaluation and benchmark design:
Although the benchmark and evaluations were significantly improved, some reviewers may remain concerned about whether the task design and reported gains truly measure deeper multimodal understanding, as opposed to benefiting from dataset filtering choices and system-level complexity.

**Reviewer Scores:**

Reviewer yfqm (initial score: 6):
Generally positive about the benchmark contribution and the multi-agent framework, while noting concerns regarding computational cost and limited technical novelty.

Reviewer bVyF (initial score: 4):
Raised concerns about missing related work, potential model bias, high computational cost for long videos, and insufficient scalability analysis. Although the rebuttal improved clarity and added supporting analyses, these concerns appear only partially resolved, and the score would likely remain below the acceptance threshold.

Reviewer 9Q2y (initial score: 4):
Raised multiple concerns, including insufficient empirical support for efficiency claims, limited citations and contextualization in the introduction, an unconvincing argument regarding data source bias, a weak connection between the Related Work and the proposed system, ambiguous baseline descriptions, limited comparative and generalization experiments, and missing ablations on key system components. While several of these points were addressed in the rebuttal, the overall assessment would likely remain below the acceptance threshold.

Reviewer e7WQ (initial score: 4):
Expressed strong skepticism regarding the nature of the contribution, viewing it primarily as system-level integration of existing components rather than a fundamentally new technical advance. Although many technical questions were addressed in the rebuttal, this core concern appears unchanged, and the score would likely remain below the acceptance threshold.

Final Recommendation:

Across reviews, there are shared concerns regarding missing or insufficient discussion of related work, limited empirical evidence for some claims, and the level of technical novelty. Although many of these issues were addressed in the rebuttal, the remaining concerns are largely conceptual and evaluative in nature. As such, the AC cannot assume that the paper would be championed by reviewers even after full discussion. Therefore, the recommendation is rejection, and the AC encourages the authors to further strengthen the paper and consider resubmission to a future venue.

---

### Decision · Program_Chairs · 2026-01-26

Reject